# Search Improves Label for Active Learning

**Alina Beygelzimer**
Yahoo Research
New York, NY
beygel@yahoo-inc.com

**Daniel Hsu**
Columbia University
New York, NY
djhsu@cs.columbia.edu

**John Langford**
Microsoft Research
New York, NY
jcl@microsoft.com

**Chicheng Zhang**
UC San Diego
La Jolla, CA
chz038@cs.ucsd.edu

## Abstract

We investigate active learning with access to two distinct oracles: LABEL (which is standard) and SEARCH (which is not). The SEARCH oracle models the situation where a human searches a database to seed or counterexample an existing solution. SEARCH is stronger than LABEL while being natural to implement in many situations. We show that an algorithm using both oracles can provide exponentially large problem-dependent improvements over LABEL alone.

## 1 Introduction

Most active learning theory is based on interacting with a LABEL oracle: An active learner observes unlabeled examples, each with a label that is initially hidden. The learner provides an unlabeled example to the oracle, and the oracle responds with the label. Using LABEL in an active learning algorithm is known to give (sometimes exponentially large) problem-dependent improvements in label complexity, even in the agnostic setting where no assumption is made about the underlying distribution [e.g., Balcan et al., 2006, Hanneke, 2007, Dasgupta et al., 2007, Hanneke, 2014].

A well-known deficiency of LABEL arises in the presence of rare classes in classification problems, frequently the case in practice [Attenberg and Provost, 2010, Simard et al., 2014]. Class imbalance may be so extreme that simply finding an example from the rare class can exhaust the labeling budget. Consider the problem of learning interval functions in $[0, 1]$. Any LABEL-only active learner needs at least $\Omega(1/\epsilon)$ LABEL queries to learn an arbitrary target interval with error at most $\epsilon$ [Dasgupta, 2005]. Given any positive example from the interval, however, the query complexity of learning intervals collapses to $O(\log(1/\epsilon))$, as we can just do a binary search for each of the end points.

A natural approach used to overcome this hurdle in practice is to search for known examples of the rare class [Attenberg and Provost, 2010, Simard et al., 2014]. Domain experts are often adept at finding examples of a class by various, often clever means. For instance, when building a hate speech filter, a simple web search can readily produce a set of positive examples. Sending a random batch of unlabeled text to LABEL is unlikely to produce any positive examples at all.

Another form of interaction common in practice is providing counterexamples to a learned predictor. When monitoring the stream filtered by the current hate speech filter, a human editor may spot a clear-cut example of hate speech that seeped through the filter. The editor, using all the search tools available to her, may even be tasked with searching for such counterexamples. The goal of the learning system is then to interactively restrict the searchable space, guiding the search process to where it is most effective.

Counterexamples can be ineffective or misleading in practice as well. Reconsidering the intervals example above, a counterexample on the boundary of an incorrect interval provides no useful information about any other examples. What is a good counterexample? What is a natural way to restrict the searchable space? How can the intervals problem be generalized?

We define a new oracle, SEARCH, that provides counterexamples to version spaces. Given a set of possible classifiers $H$ mapping unlabeled examples to labels, a *version space* $V \subseteq H$ is the subset of classifiers still under consideration by the algorithm. A *counterexample* to a version space is a labeled example which every classifier in the version space classifies incorrectly. When there is no counterexample to the version space, SEARCH returns nothing.

How can a counterexample to the version space be used? We consider a nested sequence of hypothesis classes of increasing complexity, akin to Structural Risk Minimization (SRM) in passive learning [see, e.g., Vapnik, 1982, Devroye et al., 1996]. When SEARCH produces a counterexample to the version space, it gives a proof that the current hypothesis class is too simplistic to solve the problem effectively. We show that this guided increase in hypothesis complexity results in a radically lower LABEL complexity than directly learning on the complex space. Sample complexity bounds for model selection in LABEL-only active learning were studied by Balcan et al. [2010], Hanneke [2011].

SEARCH can easily model the practice of seeding discussed earlier. If the first hypothesis class has just the constant always-negative classifier $h(x) = -1$, a seed example with label $+1$ is a counterexample to the version space. Our most basic algorithm uses SEARCH just once before using LABEL, but it is clear from inspection that multiple seeds are not harmful, and they may be helpful if they provide the proof required to operate with an appropriately complex hypothesis class.

Defining SEARCH with respect to a version space rather than a single classifier allows us to formalize "counterexample far from the boundary" in a general fashion which is compatible with the way LABEL-based active learning algorithms work.

**Related work.** The closest oracle considered in the literature is the Class Conditional Query (CCQ) [Balcan and Hanneke, 2012] oracle. A query to CCQ specifies a finite set of unlabeled examples and a label while returning an example in the subset with the specified label, if one exists.

In contrast, SEARCH has an implicit query set that is an entire region of the input space rather than a finite set. Simple searches over this large implicit domain can more plausibly discover relevant counterexamples: When building a detector for penguins in images, the input to CCQ might be a set of images and the label "penguin". Even if we are very lucky and the set happens to contain a penguin image, a search amongst image tags may fail to find it in the subset because it is not tagged appropriately. SEARCH is more likely to discover counterexamples—surely there are many images correctly tagged as having penguins.

Why is it natural to define a query region implicitly via a version space? There is a practical reason—it is a concise description of a natural region with an efficiently implementable membership filter [Beygelzimer et al., 2010, 2011, Huang et al., 2015]. (Compare this to an oracle call that has to explicitly enumerate a large set of examples. The algorithm of Balcan and Hanneke [2012] uses samples of size roughly $d\nu/\epsilon^2$.)

The use of SEARCH in this paper is also substantially different from the use of CCQ by Balcan and Hanneke [2012]. Our motivation is to use SEARCH to assist LABEL, as opposed to using SEARCH alone. This is especially useful in any setting where the cost of SEARCH is significantly higher than the cost of LABEL—we hope to avoid using SEARCH queries whenever it is possible to make progress using LABEL queries. This is consistent with how interactive learning systems are used in practice. For example, the Interactive Classification and Extraction system of Simard et al. [2014] combines LABEL with search in a production environment.

The final important distinction is that we require SEARCH to return the label of the optimal predictor in the nested sequence. For many natural sequences of hypothesis classes, the Bayes optimal classifier is eventually in the sequence, in which case it is equivalent to assuming that the label in a counterexample is the most probable one, as opposed to a randomly-drawn label from the conditional distribution (as in CCQ and LABEL).

Is this a reasonable assumption? Unlike with LABEL queries, where the labeler has no choice of what to label, here the labeler *chooses* a counterexample. If a human editor finds an unquestionable

example of hate speech that seeped through the filter, it is quite reasonable to assume that this counterexample is consistent with the Bayes optimal predictor for any sensible feature representation.

**Organization.** Section 2 formally introduces the setting. Section 3 shows that SEARCH is at least as powerful as LABEL. Section 4 shows how to use SEARCH and LABEL jointly in the realizable setting where a zero-error classifier exists in the nested sequence of hypothesis classes. Section 5 handles the agnostic setting where LABEL is subject to label noise, and shows an amortized approach to combining the two oracles with a good guarantee on the total cost.

## 2 Definitions and Setting

In active learning, there is an underlying distribution $D$ over $\mathcal{X} \times \mathcal{Y}$, where $\mathcal{X}$ is the instance space and $\mathcal{Y} := \{-1, +1\}$ is the label space. The learner can obtain independent draws from $D$, but the label is hidden unless explicitly requested through a query to the LABEL oracle. Let $D_\mathcal{X}$ denote the marginal of $D$ over $\mathcal{X}$.

We consider learning with a nested sequence of hypotheses classes $H_0 \subset H_1 \subset \cdots \subset H_k \cdots$, where $H_k \subseteq \mathcal{Y}^\mathcal{X}$ has VC dimension $d_k$. For a set of labeled examples $S \subseteq \mathcal{X} \times \mathcal{Y}$, let $H_k(S) := \{h \in H_k : \forall (x, y) \in S \,.\, h(x) = y\}$ be the set of hypotheses in $H_k$ consistent with $S$. Let $\mathrm{err}(h) := \Pr_{(x,y)\sim D}[h(x) \neq y]$ denote the error rate of a hypothesis $h$ with respect to distribution $D$, and $\mathrm{err}(h, S)$ be the error rate of $h$ on the labeled examples in $S$. Let $h_k^* = \arg\min_{h \in H_k} \mathrm{err}(h)$ breaking ties arbitrarily and let $k^* := \arg\min_{k \geq 0} \mathrm{err}(h_k^*)$ breaking ties in favor of the smallest such $k$. For simplicity, we assume the minimum is attained at some finite $k^*$. Finally, define $h^* := h_{k^*}^*$, the optimal hypothesis in the sequence of classes. The goal of the learner is to learn a hypothesis with error rate not much more than that of $h^*$.

In addition to LABEL, the learner can also query SEARCH with a version space.

---

**Oracle** $\text{SEARCH}_H(V)$ (where $H \in \{H_k\}_{k=0}^\infty$)

**input:** Set of hypotheses $V \subset H$
**output:** Labeled example $(x, h^*(x))$ s.t. $h(x) \neq h^*(x)$ for all $h \in V$, or $\perp$ if there is no such example.

---

Thus if $\text{SEARCH}_H(V)$ returns an example, this example is a *systematic* mistake made by all hypotheses in $V$. (If $V = \emptyset$, we expect SEARCH to return some example, i.e., not $\perp$.)

Our analysis is given in terms of the *disagreement coefficient* of Hanneke [2007], which has been a central parameter for analyzing active learning algorithms. Define the *region of disagreement* of a set of hypotheses $V$ as $\mathrm{Dis}(V) := \{x \in \mathcal{X} : \exists h, h' \in V \text{ s.t. } h(x) \neq h'(x)\}$. The *disagreement coefficient of $V$ at scale $r$* is $\theta_V(r) := \sup_{h \in V, r' \geq r} \Pr_{D_\mathcal{X}}[\mathrm{Dis}(\mathrm{B}_V(h, r'))]/r'$, where $\mathrm{B}_V(h, r') = \{h' \in V : \Pr_{x \sim D_\mathcal{X}}[h'(x) \neq h(x)] \leq r'\}$ is the ball of radius $r'$ around $h$.

The $\tilde{O}(\cdot)$ notation hides factors that are polylogarithmic in $1/\delta$ and quantities that do appear, where $\delta$ is the usual confidence parameter.

## 3 The Relative Power of the Two Oracles

Although SEARCH cannot always implement LABEL efficiently, it is as effective at reducing the region of disagreement. The clearest example is learning threshold classifiers $H := \{h_w : w \in [0, 1]\}$ in the realizable case, where $h_w(x) = +1$ if $w \leq x \leq 1$, and $-1$ if $0 \leq x < w$. A simple binary search with LABEL achieves an exponential improvement in query complexity over passive learning. The agreement region of any set of threshold classifiers with thresholds in $[w_{\min}, w_{\max}]$ is $[0, w_{\min}) \cup [w_{\max}, 1]$. Since SEARCH is allowed to return any counterexample in the agreement region, there is no mechanism for forcing SEARCH to return the label of a particular point we want. However, this is not needed to achieve logarithmic query complexity with SEARCH: If binary search starts with querying the label of $x \in [0, 1]$, we can query $\text{SEARCH}_H(V_x)$, where $V_x := \{h_w \in H : w < x\}$ instead. If SEARCH returns $\perp$, we know that the target $w^* \leq x$ and can safely reduce the region of disagreement to $[0, x)$. If SEARCH returns a counterexample $(x_0, -1)$ with $x_0 \geq x$, we know that $w^* > x_0$ and can reduce the region of disagreement to $(x_0, 1]$.

This observation holds more generally. In the proposition below, we assume that $\text{LABEL}(x) = h^*(x)$ for simplicity. If $\text{LABEL}(x)$ is noisy, the proposition holds for any active learning algorithm that doesn't eliminate any $h \in H : h(x) = \text{LABEL}(x)$ from the version space.

**Proposition 1.** *For any call $x \in \mathcal{X}$ to* LABEL *such that* $\text{LABEL}(x) = h^*(x)$, *we can construct a call to* SEARCH *that achieves a no lesser reduction in the region of disagreement.*

*Proof.* For any $V \subseteq H$, let $H_{\text{SEARCH}}(V)$ be the hypotheses in $H$ consistent with the output of $\text{SEARCH}_H(V)$: if $\text{SEARCH}_H(V)$ returns a counterexample $(x, y)$ to $V$, then $H_{\text{SEARCH}}(V) := \{h \in H : h(x) = y\}$; otherwise, $H_{\text{SEARCH}}(V) := V$. Let $H_{\text{LABEL}}(x) := \{h \in H : h(x) = \text{LABEL}(x)\}$. Also, let $V_x := H_{+1}(x) := \{h \in H : h(x) = +1\}$. We will show that $V_x$ is such that $H_{\text{SEARCH}}(V_x) \subseteq H_{\text{LABEL}}(x)$, and hence $\text{Dis}(H_{\text{SEARCH}}(V_x)) \subseteq \text{Dis}(H_{\text{LABEL}}(x))$.

There are two cases to consider: If $h^*(x) = +1$, then $\text{SEARCH}_H(V_x)$ returns $\bot$. In this case, $H_{\text{LABEL}}(x) = H_{\text{SEARCH}}(V_x) = H_{+1}(x)$, and we are done. If $h^*(x) = -1$, $\text{SEARCH}(V_x)$ returns a valid counterexample (possibly $(x, -1)$) in the region of agreement of $H_{+1}(x)$, eliminating all of $H_{+1}(x)$. Thus $H_{\text{SEARCH}}(V_x) \subset H \setminus H_{+1}(x) = H_{\text{LABEL}}(x)$, and the claim holds also. $\square$

As shown by the problem of learning intervals on the line, SEARCH can be exponentially more powerful than LABEL.

# 4 Realizable Case

We now turn to general active learning algorithms that combine SEARCH and LABEL. We focus on algorithms using *both* SEARCH and LABEL since LABEL is typically easier to implement than SEARCH and hence should be used where SEARCH has no significant advantage. (Whenever SEARCH is less expensive than LABEL, Section 3 suggests a transformation to a SEARCH-only algorithm.)

This section considers the realizable case, in which we assume that the hypothesis $h^* = h_{k^*}^* \in H_{k^*}$ has $\text{err}(h^*) = 0$. This means that $\text{LABEL}(x)$ returns $h^*(x)$ for any $x$ in the support of $D_{\mathcal{X}}$.

## 4.1 Combining LABEL and SEARCH

Our algorithm (shown as Algorithm 1) is called LARCH, because it combines LABEL and SEARCH. Like many selective sampling methods, LARCH uses a version space to determine its LABEL queries.

For concreteness, we use (a variant of) the algorithm of Cohn et al. [1994], denoted by CAL, as a subroutine in LARCH. The inputs to CAL are: a version space $V$, the LABEL oracle, a target error rate, and a confidence parameter; and its output is a set of labeled examples (implicitly defining a new version space). CAL is described in Appendix B; its essential properties are specified in Lemma 1.

LARCH differs from LABEL-only active learners (like CAL) by first calling SEARCH in Step 3. If SEARCH returns $\bot$, LARCH checks to see if the last call to CAL resulted in a small-enough error, halting if so in Step 6, and decreasing the allowed error rate if not in Step 8. If SEARCH instead returns a counterexample, the hypothesis class $H_k$ must be impoverished, so in Step 12, LARCH increases the complexity of the hypothesis class to the minimum complexity sufficient to correctly classify all known labeled examples in $S$. After the SEARCH, CAL is called in Step 14 to discover a sufficiently low-error (or at least low-disagreement) version space with high probability.

When LARCH advances to index $k$ (for any $k \leq k^*$), its set of labeled examples $S$ may imply a version space $H_k(S) \subseteq H_k$ that can be actively-learned more efficiently than the whole of $H_k$. In our analysis, we quantify this through the disagreement coefficient of $H_k(S)$, which may be markedly smaller than that of the full $H_k$.

The following theorem bounds the oracle query complexity of Algorithm 1 for learning with both SEARCH and LABEL in the realizable setting. The proof is in section 4.2.

**Theorem 1.** *Assume that* $\text{err}(h^*) = 0$. *For each* $k' \geq 0$, *let* $\theta_{k'}(\cdot)$ *be the disagreement coefficient of* $H_{k'}(S_{[k']})$, *where* $S_{[k']}$ *is the set of labeled examples $S$ in* LARCH *at the first time that* $k \geq k'$. *Fix any* $\epsilon, \delta \in (0, 1)$. *If* LARCH *is run with inputs hypothesis classes* $\{H_k\}_{k=0}^{\infty}$, *oracles* LABEL *and* SEARCH, *and learning parameters* $\epsilon, \delta$, *then with probability at least* $1 - \delta$: LARCH *halts after at most* $k^* + \log_2(1/\epsilon)$ *for-loop iterations and returns a classifier with error rate at most* $\epsilon$; *furthermore,*

---

**Algorithm 1** LARCH

---

**input:** Nested hypothesis classes $H_0 \subset H_1 \subset \cdots$; oracles LABEL and SEARCH; learning parame-
    ters $\epsilon, \delta \in (0, 1)$

1: **initialize** $S \leftarrow \emptyset$, (index) $k \leftarrow 0, \ell \leftarrow 0$
2: **for** $i = 1, 2, \ldots$ **do**
3:     $e \leftarrow \text{SEARCH}_{H_k}(H_k(S))$
4:     **if** $e = \perp$ **then**                                           # no counterexample found
5:       **if** $2^{-\ell} \leq \epsilon$ **then**
6:         **return** any $h \in H_k(S)$
7:       **else**
8:         $\ell \leftarrow \ell + 1$
9:       **end if**
10:    **else**                                                # counterexample found
11:       $S \leftarrow S \cup \{e\}$
12:       $k \leftarrow \min\{k' : H_{k'}(S) \neq \emptyset\}$
13:    **end if**
14:    $S \leftarrow S \cup \text{CAL}(H_k(S), \text{LABEL}, 2^{-\ell}, \delta/(i^2 + i))$
15: **end for**

---

it draws at most $\tilde{O}(k^* d_{k^*}/\epsilon)$ unlabeled examples from $D_{\mathcal{X}}$, makes at most $k^* + \log_2(1/\epsilon)$ queries to SEARCH, and at most $\tilde{O}\left((k^* + \log(1/\epsilon)) \cdot (\max_{k' \leq k^*} \theta_{k'}(\epsilon)) \cdot d_{k^*} \cdot \log^2(1/\epsilon)\right)$ queries to LABEL.

**Union-of-intervals example.** We now show an implication of Theorem 1 in the case where the target hypothesis $h^*$ is the union of non-trivial intervals in $\mathcal{X} := [0, 1]$, assuming that $D_{\mathcal{X}}$ is uniform. For $k \geq 0$, let $H_k$ be the hypothesis class of the union of up to $k$ intervals in $[0, 1]$ with $H_0$ containing only the always-negative hypothesis. (Thus, $h^*$ is the union of $k^*$ non-empty intervals.) The disagreement coefficient of $H_1$ is $\Omega(1/\epsilon)$, and hence LABEL-only active learners like CAL are not very effective at learning with such classes. However, the first SEARCH query by LARCH provides a counterexample to $H_0$, which must be a positive example $(x_1, +1)$. Hence, $H_1(S_{[1]})$ (where $S_{[1]}$ is defined in Theorem 1) is the class of intervals that contain $x_1$ with disagreement coefficient $\theta_1 \leq 4$.

Now consider the inductive case. Just before LARCH advances its index to a value $k$ (for any $k \leq k^*$), SEARCH returns a counterexample $(x, h^*(x))$ to the version space; every hypothesis in this version space (which could be empty) is a union of fewer than $k$ intervals. If the version space is empty, then $S$ must already contain positive examples from at least $k$ different intervals in $h^*$ and at least $k - 1$ negative examples separating them. If the version space is not empty, then the point $x$ is either a positive example belonging to a previously uncovered interval in $h^*$ or a negative example splitting an existing interval. In either case, $S_{[k]}$ contains positive examples from at least $k$ distinct intervals separated by at least $k - 1$ negative examples. The disagreement coefficient of the set of unions of $k$ intervals consistent with $S_{[k]}$ is at most $4k$, independent of $\epsilon$.

The VC dimension of $H_k$ is $O(k)$, so Theorem 1 implies that with high probability, LARCH makes at most $k^* + \log(1/\epsilon)$ queries to SEARCH and $\tilde{O}((k^*)^3 \log(1/\epsilon) + (k^*)^2 \log^3(1/\epsilon))$ queries to LABEL.

## 4.2 Proof of Theorem 1

The proof of Theorem 1 uses the following lemma regarding the CAL subroutine, proved in Appendix B. It is similar to a result of Hanneke [2011], but an important difference here is that the input version space $V$ is not assumed to contain $h^*$.

**Lemma 1.** *Assume* $\text{LABEL}(x) = h^*(x)$ *for every $x$ in the support of $D_{\mathcal{X}}$. For any hypothesis set $V \subseteq \mathcal{Y}^{\mathcal{X}}$ with VC dimension $d < \infty$, and any $\epsilon, \delta \in (0, 1)$, the following holds with probability at least $1 - \delta$.* $\text{CAL}(V, \text{LABEL}, \epsilon, \delta)$ *returns labeled examples $T \subseteq \{(x, h^*(x)) : x \in \mathcal{X}\}$ such that for any $h$ in $V(T)$, $\Pr_{(x,y)\sim D}[h(x) \neq y \wedge x \in \text{Dis}(V(T))] \leq \epsilon$; furthermore, it draws at most $\tilde{O}(d/\epsilon)$ unlabeled examples from $D_{\mathcal{X}}$, and makes at most $\tilde{O}\left(\theta_V(\epsilon) \cdot d \cdot \log^2(1/\epsilon)\right)$ queries to LABEL.*

We now prove Theorem 1. By Lemma 1 and a union bound, there is an event with probability at least $1 - \sum_{i \geq 1} \delta/(i^2 + i) \geq 1 - \delta$ such that each call to CAL made by LARCH satisfies the high-probability guarantee from Lemma 1. We henceforth condition on this event.

We first establish the guarantee on the error rate of a hypothesis returned by LARCH. By the assumed properties of LABEL and SEARCH, and the properties of CAL from Lemma 1, the labeled examples $S$ in LARCH are always consistent with $h^*$. Moreover, the return property of CAL implies that at the end of any loop iteration, with the present values of $S$, $k$, and $\ell$, we have $\Pr_{(x,y)\sim D}[h(x) \neq y \wedge x \in \mathrm{Dis}(H_k(S))] \leq 2^{-\ell}$ for all $h \in H_k(S)$. (The same holds trivially before the first loop iteration.) Therefore, if LARCH halts and returns a hypothesis $h \in H_k(S)$, then there is no counterexample to $H_k(S)$, and $\Pr_{(x,y)\sim D}[h(x) \neq y \wedge x \in \mathrm{Dis}(H_k(S))] \leq \epsilon$. These consequences and the law of total probability imply $\mathrm{err}(h) = \Pr_{(x,y)\sim D}[h(x) \neq y \wedge x \in \mathrm{Dis}(H_k(S))] \leq \epsilon$.

We next consider the number of for-loop iterations executed by LARCH. Let $S_i$, $k_i$, and $\ell_i$ be, respectively, the values of $S$, $k$, and $\ell$ at the start of the $i$-th for-loop iteration in LARCH. We claim that if LARCH does not halt in the $i$-th iteration, then one of $k$ and $\ell$ is incremented by at least one. Clearly, if there is no counterexample to $H_{k_i}(S_i)$ and $2^{-\ell_i} > \epsilon$, then $\ell$ is incremented by one (Step 8). If, instead, there is a counterexample $(x,y)$, then $H_{k_i}(S_i \cup \{(x,y)\}) = \emptyset$, and hence $k$ is incremented to some index larger than $k_i$ (Step 12). This proves that $k_{i+1} + \ell_{i+1} \geq k_i + \ell_i + 1$. We also have $k_i \leq k^*$, since $h^* \in H_{k^*}$ is consistent with $S$, and $\ell_i \leq \log_2(1/\epsilon)$, as long as LARCH does not halt in for-loop iteration $i$. So the total number of for-loop iterations is at most $k^* + \log_2(1/\epsilon)$. Together with Lemma 1, this bounds the number of unlabeled examples drawn from $D_{\mathcal{X}}$.

Finally, we bound the number of queries to SEARCH and LABEL. The number of queries to SEARCH is the same as the number of for-loop iterations—this is at most $k^* + \log_2(1/\epsilon)$. By Lemma 1 and the fact that $V(S' \cup S'') \subseteq V(S')$ for any hypothesis space $V$ and sets of labeled examples $S', S''$, the number of LABEL queries made by CAL in the $i$-th for-loop iteration is at most $\tilde{O}(\theta_{k_i}(\epsilon) \cdot d_{k_i} \cdot \ell_i^2 \cdot \mathrm{polylog}(i))$. The claimed bound on the number of LABEL queries made by LARCH now readily follows by taking a max over $i$, and using the facts that $i \leq k^*$ and $d_{k'} \leq d_{k^*}$ for all $k' \leq k$. $\qquad\square$

### 4.3 An Improved Algorithm

LARCH is somewhat conservative in its use of SEARCH, interleaving just one SEARCH query between sequences of LABEL queries (from CAL). Often, it is advantageous to advance to higher complexity hypothesis classes quickly, as long as there is justification to do so. Counterexamples from SEARCH provide such justification, and a $\perp$ result from SEARCH also provides useful feedback about the current version space: outside of its disagreement region, the version space is in complete agreement with $h^*$ (even if the version space does not contain $h^*$). Based on these observations, we propose an improved algorithm for the realizable setting, which we call SEABEL. Due to space limitations, we present it in Appendix C. We prove the following performance guarantee for SEABEL.

**Theorem 2.** *Assume that* $\mathrm{err}(h^*) = 0$. *Let* $\theta_k(\cdot)$ *denote the disagreement coefficient of* $V_i^{k_i}$ *at the first iteration* $i$ *in* SEABEL *where* $k_i \geq k$. *Fix any* $\epsilon, \delta \in (0,1)$. *If* SEABEL *is run with inputs hypothesis classes* $\{H_k\}_{k=0}^{\infty}$, *oracles* SEARCH *and* LABEL, *and learning parameters* $\epsilon, \delta \in (0,1)$, *then with probability* $1 - \delta$: SEABEL *halts and returns a classifier with error rate at most* $\epsilon$; *furthermore, it draws at most* $\tilde{O}((d_{k^*} + \log k^*)/\epsilon)$ *unlabeled examples from* $D_{\mathcal{X}}$, *makes at most* $k^* + O\left(\log(d_{k^*}/\epsilon) + \log \log k^*\right)$ *queries to* SEARCH, *and at most* $\tilde{O}\left(\max_{k \leq k^*} \theta_k(2\epsilon) \cdot (d_{k^*} \log^2(1/\epsilon) + \log k^*)\right)$ *queries to* LABEL.

It is not generally possible to directly compare Theorems 1 and 2 on account of the algorithm-dependent disagreement coefficient bounds. However, in cases where these disagreement coefficients are comparable (as in the union-of-intervals example), the SEARCH complexity in Theorem 2 is slightly higher (by additive log terms), but the LABEL complexity is smaller than that from Theorem 1 by roughly a factor of $k^*$. For the union-of-intervals example, SEABEL would learn target union of $k^*$ intervals with $k^* + O(\log(k^*/\epsilon))$ queries to SEARCH and $\tilde{O}((k^*)^2 \log^2(1/\epsilon))$ queries to LABEL.

## 5 Non-Realizable Case

In this section, we consider the case where the optimal hypothesis $h^*$ may have non-zero error rate, i.e., the non-realizable (or agnostic) setting. In this case, the algorithm LARCH, which was designed for the realizable setting, is no longer applicable. First, examples obtained by LABEL and SEARCH are of different quality: those returned by SEARCH always agree with $h^*$, whereas the labels given by LABEL need not agree with $h^*$. Moreover, the version spaces (even when $k = k^*$) as defined by LARCH may always be empty due to the noisy labels.

Another complication arises in our SRM setting that differentiates it from the usual agnostic active learning setting. When working with a specific hypothesis class $H_k$ in the nested sequence, we may observe high error rates because (i) the finite sample error is too high (but additional labeled examples could reduce it), or (ii) the current hypothesis class $H_k$ is impoverished. In case (ii), the best hypothesis in $H_k$ may have a much larger error rate than $h^*$, and hence lower bounds [Kääriäinen, 2006] imply that active learning on $H_k$ instead of $H_{k^*}$ may be substantially more difficult.

These difficulties in the SRM setting are circumvented by an algorithm that adaptively estimates the error of $h^*$. The algorithm, A-LARCH (Algorithm 5), is presented in Appendix D.

**Theorem 3.** *Assume* $\mathrm{err}(h^*) = \nu$. *Let* $\theta_k(\cdot)$ *denote the disagreement coefficient of* $V_i^{k_i}$ *at the first iteration* $i$ *in* A-LARCH *where* $k_i \geq k$. *Fix any* $\epsilon, \delta \in (0,1)$. *If* A-LARCH *is run with inputs hypothesis classes* $\{H_k\}_{k=0}^{\infty}$, *oracles* SEARCH *and* LABEL, *learning parameter* $\delta$, *and unlabeled example budget* $\tilde{O}((d_{k^*} + \log k^*)(\nu + \epsilon)/\epsilon^2)$, *then with probability* $1 - \delta$: A-LARCH *returns a classifier with error rate* $\leq \nu + \epsilon$; *it makes at most* $k^* + O\left(\log(d_{k^*}/\epsilon) + \log \log k^*\right)$ *queries to* SEARCH, *and* $\tilde{O}\left(\max_{k \leq k^*} \theta_k(2\nu + 2\epsilon) \cdot (d_{k^*} \log^2(1/\epsilon) + \log k^*) \cdot (1 + \nu^2/\epsilon^2)\right)$ *queries to* LABEL.

The proof is in Appendix D. The LABEL query complexity is at least a factor of $k^*$ better than that in Hanneke [2011], and sometimes exponentially better thanks to the reduced disagreement coefficient of the version space when consistency constraints are incorporated.

## 5.1 AA-LARCH: an Opportunistic Anytime Algorithm

In many practical scenarios, termination conditions based on quantities like a target excess error rate $\epsilon$ are undesirable. The target $\epsilon$ is unknown, and we instead prefer an algorithm that performs as well as possible until a cost budget is exhausted. Fortunately, when the primary cost being considered are LABEL queries, there are many LABEL-only active learning algorithms that readily work in such an "anytime" setting [see, e.g., Dasgupta et al., 2007, Hanneke, 2014].

The situation is more complicated when we consider both SEARCH and LABEL: we can often make substantially more progress with SEARCH queries than with LABEL queries (as the error rate of the best hypothesis in $H_{k'}$ for $k' > k$ can be far lower than in $H_k$). AA-LARCH (Algorithm 2) shows that although these queries come at a higher cost, the cost can be amortized.

AA-LARCH relies on several subroutines: SAMPLE-AND-LABEL, ERROR-CHECK, PRUNE-VERSION-SPACE and UPGRADE-VERSION-SPACE (Algorithms 6, 7, 8, and 9). The detailed descriptions are deferred to Appendix E. SAMPLE-AND-LABEL performs standard disagreement-based selective sampling using oracle LABEL; labels of examples in the disagreement region are queried, otherwise inferred. PRUNE-VERSION-SPACE prunes the version space given the labeled examples collected, based on standard generalization error bounds. ERROR-CHECK checks if the best hypothesis in the version space has large error; SEARCH is used to find a systematic mistake for the version space; if either event happens, AA-LARCH calls UPGRADE-VERSION-SPACE to increase $k$, the level of our working hypothesis class.

**Theorem 4.** *Assume* $\mathrm{err}(h^*) = \nu$. *Let* $\theta_{k'}(\cdot)$ *denote the disagreement coefficient of* $V_i$ *at the first iteration* $i$ *after which* $k \geq k'$. *Fix any* $\epsilon \in (0,1)$. *Let* $n_\epsilon = \tilde{O}(\max_{k \leq k^*} \theta_k(2\nu + 2\epsilon)d_{k^*}(1 + \nu^2/\epsilon^2))$ *and define* $C_\epsilon = 2(n_\epsilon + k^*\tau)$. *Run Algorithm 2 with a nested sequence of hypotheses* $\{H_k\}_{k=0}^{\infty}$, *oracles* LABEL *and* SEARCH, *confidence parameter* $\delta$, *cost ratio* $\tau \geq 1$, *and upper bound* $N = \tilde{O}(d_{k^*}/\epsilon^2)$. *If the cost spent is at least* $C_\epsilon$, *then with probability* $1 - \delta$, *the current hypothesis* $\tilde{h}$ *has error at most* $\nu + \epsilon$.

The proof is in Appendix E. A comparison to Theorem 3 shows that AA-LARCH is adaptive: for any cost complexity $C$, the excess error rate $\epsilon$ is roughly at most twice that achieved by A-LARCH.

## 6 Discussion

The SEARCH oracle captures a powerful form of interaction that is useful for machine learning. Our theoretical analyses of LARCH and variants demonstrate that SEARCH can substantially improve LABEL-based active learners, while being plausibly cheaper to implement than oracles like CCQ.

---
**Algorithm 2** AA-LARCH
---
**input:** Nested hypothesis set $H_0 \subseteq H_1 \subseteq \cdots$; oracles LABEL and SEARCH; learning parameter
$\quad$ $\delta \in (0, 1)$; SEARCH-to-LABEL cost ratio $\tau$, dataset size upper bound $N$.
**output:** hypothesis $\tilde{h}$.
$\quad$ 1: Initialize: consistency constraints $S \leftarrow \emptyset$, counter $c \leftarrow 0$, $k \leftarrow 0$, verified labeled dataset $\tilde{L} \leftarrow \emptyset$,
$\qquad$ working labeled dataset $L_0 \leftarrow \emptyset$, unlabeled examples processed $i \leftarrow 0$, $V_i \leftarrow H_k(S)$.
$\quad$ 2: **loop**
$\quad$ 3: $\quad$ Reset counter $c \leftarrow 0$.
$\quad$ 4: $\quad$ **repeat**
$\quad$ 5: $\qquad$ **if** ERROR-CHECK$(V_i, L_i, \delta_i)$ **then**
$\quad$ 6: $\qquad\quad$ $(k, S, V_i) \leftarrow$ UPGRADE-VERSION-SPACE$(k, S, \emptyset)$
$\quad$ 7: $\qquad\quad$ $V_i \leftarrow$ PRUNE-VERSION-SPACE$(V_i, \tilde{L}, \delta_i)$
$\quad$ 8: $\qquad\quad$ $L_i \leftarrow \tilde{L}$
$\quad$ 9: $\qquad\quad$ **continue loop**
$\quad$ 10: $\qquad$ **end if**
$\quad$ 11: $\qquad$ $i \leftarrow i + 1$
$\quad$ 12: $\qquad$ $(L_i, c) \leftarrow$ SAMPLE-AND-LABEL$(V_{i-1}, \text{LABEL}, L_{i-1}, c)$
$\quad$ 13: $\qquad$ $V_i \leftarrow$ PRUNE-VERSION-SPACE$(V_{i-1}, L_i, \delta_i)$
$\quad$ 14: $\quad$ **until** $c = \tau$ **or** $l_i = N$
$\quad$ 15: $\quad$ $e \leftarrow$ SEARCH$_{H_k}(V_i)$
$\quad$ 16: $\quad$ **if** $e \neq \perp$ **then**
$\quad$ 17: $\qquad$ $(k, S, V_i) \leftarrow$ UPGRADE-VERSION-SPACE$(k, S, \{e\})$
$\quad$ 18: $\qquad$ $V_i \leftarrow$ PRUNE-VERSION-SPACE$(V_i, \tilde{L}, \delta_i)$
$\quad$ 19: $\qquad$ $L_i \leftarrow \tilde{L}$
$\quad$ 20: $\quad$ **else**
$\quad$ 21: $\qquad$ Update verified dataset $\tilde{L} \leftarrow L_i$.
$\quad$ 22: $\qquad$ Store temporary solution $\tilde{h} = \arg\min_{h' \in V_i} \text{err}(h', \tilde{L})$.
$\quad$ 23: $\quad$ **end if**
$\quad$ 24: **end loop**
---

Are there examples where CCQ is substantially more powerful than SEARCH? This is a key question, because a good active learning system should use minimally powerful oracles. Another key question is: Can the benefits of SEARCH be provided in a computationally efficient general purpose manner?

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
