[Supplementary Material · label_and_search_long.pdf]

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

14:     **until** $c = \tau$ **or** $l_i = N$
15:     $e \leftarrow$ SEARCH$_{H_k}(V_i)$
16:     **if** $e \neq \perp$ **then**
17:       $(k, S, V_i) \leftarrow$ UPGRADE-VERSION-SPACE$(k, S, \{e\})$
18:       $V_i \leftarrow$ PRUNE-VERSION-SPACE$(V_i, \tilde{L}, \delta_i)$
19:       $L_i \leftarrow \tilde{L}$
20:     **else**
21:       Update verified dataset $\tilde{L} \leftarrow L_i$.
22:       Store temporary solution $\tilde{h} = \arg\min_{h' \in V_i} \text{err}(h', \tilde{L})$.
23:     **end if**
24: **end loop**
---

Are there examples where CCQ is substantially more powerful than SEARCH? This is a key question, because a good active learning system should use minimally powerful oracles. Another key question is: Can the benefits of SEARCH be provided in a computationally efficient general purpose manner?

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

# A Basic Facts and Notations Used in Proofs

## A.1 Concentration Inequalities

**Lemma 2** (Bernstein's Inequality). *Let $X_1, \ldots, X_n$ be independent zero-mean random variables. Suppose that $|X_i| \leq M$ almost surely. Then for all positive $t$,*

$$\Pr\left[\sum_{i=1}^{n} X_i > t\right] \leq \exp\left(-\frac{t^2/2}{\sum_{j=1}^{n} \mathrm{E}[X_j^2] + Mt/3}\right).$$

**Lemma 3.** *Let $Z_1, \ldots, Z_n$ be independent Bernoulli random variables with mean $p$. Let $\bar{Z} = \frac{1}{n}\sum_{i=1}^{n} Z_i$. Then with probability $1 - \delta$,*

$$\bar{Z} \leq p + \sqrt{\frac{2p\ln(1/\delta)}{n}} + \frac{2\ln(1/\delta)}{3n}.$$

*Proof.* Let $X_i = Z_i - p$ for all $i$, note that $|X_i| \leq 1$. The lemma follows from Bernstein's Inequality and algebra. $\square$

**Lemma 4** (Freedman's Inequality). *Let $X_1, \ldots, X_n$ be a martingale difference sequence, and $|X_i| \leq M$ almost surely. Let V be the sum of the conditional variances, i.e.*

$$V = \sum_{i=1}^{n} \mathrm{E}[X_i^2 | X_1, \ldots . X_{i-1}]$$

*Then, for every $t, v > 0$,*

$$\Pr\left[\sum_{i=1}^{n} X_i > t \text{ and } V \leq v\right] \leq \exp\left(-\frac{t^2/2}{v + Mt/3}\right).$$

**Lemma 5.** *Let $Z_1, \ldots, Z_n$ be a sequence of Bernoulli random variables, where $\mathrm{E}[Z_i | Z_1, \ldots, Z_{i-1}] = p_i$. Then, for every $\delta > 0$, with probability $1 - \delta$:*

$$\sum_{i=1}^{n} Z_i \leq 2v_n + \sqrt{4v_n \ln\frac{\log 4n}{\delta}} + \frac{2}{3}\ln\frac{\log 4n}{\delta}.$$

*where $v_n = \max(\sum_{i=1}^{n} p_i, 1)$.*

*Proof.* Let $X_i = Z_i - p_i$ for all $i$, note that $\{X_i\}$ is a martingale difference sequence and $|X_i| \leq 1$. From Freedman's Inequality and algebra, for any $v$,

$$\Pr\left[\frac{1}{n}\sum_{i=1}^{n} Z_i > v + \sqrt{\frac{2v\ln\frac{\log 4n}{\delta}}{n}} + \frac{2\ln\frac{\log 4n}{\delta}}{3n} \text{ and } \sum_{i=1}^{n} p_i \leq v\right] \leq \frac{\delta}{\log n + 2}.$$

The proof follows by taking union bound over $v = 2^i, i = 0, 1, \ldots, \lceil\log n\rceil$. $\square$

Define

$$\phi(d, m, \delta) := \frac{1}{m}\left(d\log em^2 + \log\frac{2}{\delta}\right). \tag{1}$$

**Theorem 5** (Vapnik and Chervonenkis, 1971). *Let $\mathcal{F}$ be a family of functions $f: \mathcal{Z} \to \{0, 1\}$ on a domain $\mathcal{Z}$ with VC dimension at most $d$, and let $P$ be a distribution on $\mathcal{Z}$. Let $P_n$ denote the empirical measure from an iid sample of size $n$ from $P$. For any $\delta \in (0, 1)$, with probability at least $1 - \delta$, for all $f \in \mathcal{F}$,*

$$-\min\left\{\varepsilon + \sqrt{Pf\varepsilon}, \sqrt{P_nf\varepsilon}\right\} \leq Pf - P_nf \leq \min\left\{\varepsilon + \sqrt{P_nf\varepsilon}, \sqrt{Pf\varepsilon}\right\}$$

*where $\varepsilon := \phi(d, n, \delta)$.*

## A.2 Notations

For convenience, we define

$$\sigma(d, m, \delta) := \phi(d, m, \delta/3), \tag{2}$$

as we will often split the overall allowed failure probability $\delta$ across two or three separate events.

Because we apply the deviation inequalities to the hypothesis classes from $\{H_k\}_{k=0}^{\infty}$, we also define:

$$\sigma_k(m, \delta) := \sigma(d_k, m, \delta), \tag{3}$$

where $d_k$ is the VC dimension of $H_k$. We have the following simple fact.

**Fact 1.**

$$\sigma\left(d, m, \frac{\delta}{2 \log m (\log m + 1)}\right) \geq \epsilon \implies m \leq \frac{64}{\epsilon}\left(d \log \frac{512}{\epsilon} + \log \frac{24}{\delta}\right).$$

For integers $i \geq 1$ and $k \geq 0$, define

$$\delta_i := \frac{\delta}{i(i+1)}, \qquad \delta_{i,k} := \frac{\delta_i}{(k+1)(k+2)}$$

Note that $\sum_{i=1}^{\infty} \delta_i = \delta$ and $\sum_{k=0}^{\infty} \delta_{i,k} = \delta_i$.

Finally, for any distribution $\tilde{D}$ over $\mathcal{X} \times \mathcal{Y}$ and any hypothesis $h \colon \mathcal{X} \to \mathcal{Y}$, we use $\mathrm{err}(h, \tilde{D}) := \Pr_{(x,y)\sim\tilde{D}}[h(x) \neq y]$ to denote the probability with respect to $\tilde{D}$ that $h$ makes a classification error.

# B  Active Learning Algorithm CAL

In this section, we describe and analyze a variant of the LABEL-only active learning algorithm of Cohn et al. [1994], which we refer to as CAL. Note that Hanneke [2011] provides a label complexity analysis of CAL in terms of the disagreement coefficient under the assumption that the LABEL oracle is consistent with some hypothesis in the hypothesis class used by CAL. We cannot use that analysis because we call CAL as a subroutine in LARCH with sets of hypotheses $V$ that do not necessarily contain the optimal hypothesis $h^*$.

## B.1  Description of CAL

CAL takes as input a set of hypotheses $V$, the LABEL oracle (which always returns $h^*(x)$ when queried with a point $x$), and learning parameters $\epsilon, \delta \in (0, 1)$.

The pseudocode for CAL is given in Algorithm 3 below, where we use the notation

$$U_{\leq i} := \bigcup_{j=1}^{i} U_j$$

for any sequence of sets $(U_j)_{j\in\mathbb{N}}$.

## B.2  Proof of Lemma 1

We now give the proof of Lemma 1.

Let $V_0 := V$ and $V_i := V(T_{\leq i})$ for each $i \geq 1$. Clearly $V_0 \supseteq V_1 \supseteq \cdots$, and hence $\mathrm{Dis}(V_0) \supseteq \mathrm{Dis}(V_1) \supseteq \cdots$ as well.

Let $E_i$ be the event in which the following hold:

1. If CAL executes iteration $i$, then every $h \in V_i$ satisfies

$$\Pr_{x\sim D_{\mathcal{X}}}[h(x) \neq h^*(x) \ \wedge \ x \in \mathrm{Dis}(V_i)] \leq \phi(d, 2^i, \delta_i/2).$$

**Algorithm 3** CAL

---

**input:** Hypothesis set $V$ with VC dimension $\leq d$; oracle LABEL; learning parameters $\epsilon, \delta \in (0,1)$
**output:** Labeled examples $T$
 1: **for** $i = 1, 2, \ldots$ **do**
 2:     $T_i \leftarrow \emptyset$
 3:     **for** $j = 1, 2, \ldots, 2^i$ **do**
 4:         $x_{i,j} \leftarrow$ independent draw from $D_{\mathcal{X}}$ (the corresponding label is hidden)
 5:         **if** $x_{i,j} \in \mathrm{Dis}(V(T_{\leq i-1}))$ **then**
 6:             $T_i \leftarrow T_i \cup \{(x_{i,j}, \mathrm{LABEL}(x_{i,j}))\}$
 7:         **end if**
 8:     **end for**
 9:     **if** $\phi(d, 2^i, \delta_i/2) \leq \epsilon$ or $V(T_{\leq i}) = \emptyset$ **then**
10:         **return** $T_{\leq i}$
11:     **end if**
12: **end for**

---

2. If CAL executes iteration $i$, then the number of LABEL queries in iteration $i$ is at most

$$2^i \mu_i + O\left(\sqrt{2^i \mu_i \log(2/\delta_i)} + \log(2/\delta_i)\right),$$

where

$$\mu_i := \theta_{V_{i-1}}(\epsilon) \cdot 2\phi(d, 2^{i-1}, \delta_{i-1}/2).$$

We claim that $E_0 \cap E_1 \cap \cdots \cap E_i$ holds with probability at least $1 - \sum_{i'=1}^{i} \delta_{i'} \geq 1 - \delta$. The proof is by induction. The base case is trivial, as $E_0$ holds deterministically. For the inductive case, we just have to show that $\Pr(E_i \mid E_0 \cap E_1 \cap \cdots \cap E_{i-1}) \geq 1 - \delta_i$.

Condition on the event $E_0 \cap E_1 \cap \cdots \cap E_{i-1}$. Suppose CAL executes iteration $i$. For all $x \notin \mathrm{Dis}(V_{i-1})$, let $V_{i-1}(x)$ denote the label assigned by every $h \in V_{i-1}$ to $x$. Define

$$\hat{S}_i := \left\{(x_{i,j}, \hat{y}_{i,j}) : j \in \{1, 2, \ldots, 2^i\}, \ x_{i,j} \notin \mathrm{Dis}(V_{i-1}), \ \hat{y}_{i,j} = V_{i-1}(x_{i,j})\right\}.$$

Observe that $\hat{S}_i \cup T_i$ is an iid sample of size $2^i$ from a distribution (which we call $D_{i-1}$) over labeled examples $(x, y)$, where $x \sim D_{\mathcal{X}}$ and $y$ is given by

$$y := \begin{cases} V_{i-1}(x) & \text{if } x \notin \mathrm{Dis}(V_{i-1}), \\ h^*(x) & \text{if } x \in \mathrm{Dis}(V_{i-1}). \end{cases}$$

In fact, for any $h \in V_{i-1}$, we have

$$\mathrm{err}_{D_{i-1}}(h) = \Pr_{(x,y)\sim D_{i-1}}[h(x) \neq y] = \Pr_{x\sim D_{\mathcal{X}}}[h(x) \neq h^*(x) \wedge x \in \mathrm{Dis}(V_{i-1})]. \quad (4)$$

The VC inequality (Theorem 5) implies that, with probability at least $1 - \delta_i/2$,

$$\forall h \in V \centerdot \left(\mathrm{err}(h, \hat{S}_i \cup T_i) = 0 \implies \mathrm{err}_{D_{i-1}}(h) \leq \phi(d, 2^i, \delta_i/2)\right). \quad (5)$$

Consider any $h \in V_i$. We have $\mathrm{err}(h, T_i) = 0$ by definition of $V_i$. We also have $\mathrm{err}(h, \hat{S}_i) = 0$ since $h \in V_i \subseteq V_{i-1}$. So in the event that (5) holds, we have

$$\Pr_{x\sim D_{\mathcal{X}}}[h(x) \neq h^*(x) \wedge x \in \mathrm{Dis}(V_i)] \leq \Pr_{x\sim D_{\mathcal{X}}}[h(x) \neq h^*(x) \wedge x \in \mathrm{Dis}(V_{i-1})]$$

$$= \mathrm{err}_{D_{i-1}}(h)$$

$$\leq \phi(d, 2^i, \delta_i/2),$$

where the first inequality follows because $\mathrm{Dis}(V_i) \subseteq \mathrm{Dis}(V_{i-1})$, and the equality follows from (4).

Now we prove the LABEL query bound.

**Claim 1.** *On event $E_{i-1}$ for every $h, h' \in V_{i-1}$,*

$$\Pr_{x\sim D_{\mathcal{X}}}[h(x) \neq h'(x)] \leq 2\phi(d, 2^{i-1}, \delta_{i-1}/2)$$

*Proof.* On event $E_{i-1}$, every $h \in V_{i-1}$ satisfies
$$\Pr_{x \sim D_{\mathcal{X}}}[h(x) \neq h^*(x), x \in \text{Dis}(V_{i-1})] \leq \phi(d, 2^{i-1}, \delta_{i-1}/2).$$
Therefore, for any $h, h' \in V_{i-1}$, we have
$$\begin{aligned}
\Pr_{x \sim D_{\mathcal{X}}}[h(x) \neq h'(x)] &= \Pr_{x \sim D_{\mathcal{X}}}[h(x) \neq h'(x), x \in \text{Dis}(V_{i-1})] \\
&\leq \Pr_{x \sim D_{\mathcal{X}}}[h(x) \neq h^*(x), x \in \text{Dis}(V_{i-1})] \\
&\quad + \Pr_{x \sim D_{\mathcal{X}}}[h'(x) \neq h^*(x), x \in \text{Dis}(V_{i-1})] \\
&\leq 2\phi(d, 2^{i-1}, \delta_{i-1}/2). \qquad \square
\end{aligned}$$

Since CAL does not halt before iteration $i$, we have $2\phi(d, 2^{i-1}, \delta_{i-1}/2) \geq \epsilon$, and hence the above claim and the definition of the disagreement coefficient imply
$$\Pr_{x \sim D_{\mathcal{X}}}[x \in \text{Dis}(V_{i-1})] \leq \theta_{V_{i-1}}(\epsilon) \cdot 2\phi(d, 2^{i-1}, \delta_{i-1}/2) = \mu_i.$$

Therefore, $\mu_i$ is an upper bound on the probability that LABEL is queried on $x_{i,j}$, for each $j = 1, 2, \ldots, 2^i$. By Lemma 3, the number of queries to LABEL is at most
$$2^i \mu_i + O\left(\sqrt{2^i \mu_i \log(2/\delta_i)} + \log(2/\delta_i)\right).$$
with probability at least $1 - \delta_i/2$. We conclude by a union bound that $\Pr(E_i \mid E_0 \cap E_1 \cap \cdots \cap E_{i-1}) \geq 1 - \delta_i$ as required.

We now show that in the event $E_0 \cap E_1 \cap \cdots$, which holds with probability at least $1 - \delta$, the required consequences from Lemma 1 are satisfied. The definition of $\phi$ from (1) and the halting condition in CAL imply that the number of iterations $I$ executed by CAL satisfies
$$\sigma(d, 2^{I-1}, \delta_{I-1}/2) \geq \epsilon.$$
Thus by Fact 1,
$$2^I \leq O\left(\frac{1}{\epsilon}\left(d \log \frac{1}{\epsilon} + \log \frac{1}{\delta}\right)\right),$$
which immediately gives the required bound on the number of unlabeled points drawn from $D_{\mathcal{X}}$. Moreover, $I$ can be bounded as
$$I = O\left(\log(d/\epsilon) + \log\log(1/\delta)\right).$$
Therefore, in the event $E_0 \cap E_1 \cap \cdots \cap E_I$, CAL returns a set of labeled examples $T := T_{\leq I}$ in which every $h \in V(T)$ satisfies
$$\Pr_{x \sim D_{\mathcal{X}}}[h(x) \neq h^*(x) \wedge x \in \text{Dis}(V(T))] \leq \epsilon,$$
and the number of LABEL queries is bounded by
$$\begin{aligned}
&\sum_{i=1}^{I}\left(2^i \mu_i + O\left(\sqrt{2^i \mu_i \log(2/\delta_i)} + \log(2/\delta_i)\right)\right) \\
&= \sum_{i=1}^{I} O\left(2^i \cdot \left(\theta_{V_{i-1}}(\epsilon)\frac{d \log 2^i + \log(2/\delta_i)}{2^i}\right) + \log(2/\delta_i)\right) \\
&= \sum_{i=1}^{I} O\left(\theta_{V_{i-1}}(\epsilon) \cdot \left(d \cdot i + \log(1/\delta)\right)\right) \\
&= O\left(\theta_V(\epsilon) \cdot \left(d \cdot \left(\log(d/\epsilon) + \log\log(1/\delta)\right)^2 + \left(\log(d/\epsilon) + \log\log(1/\delta)\right) \cdot \log(1/\delta)\right)\right) \\
&= \tilde{O}\left(\theta_V(\epsilon) \cdot d \cdot \log^2(1/\epsilon)\right)
\end{aligned}$$
as claimed. $\qquad \square$

---

**Algorithm 4** SEABEL

---

**input:** Nested hypothesis classes $H_0 \subseteq H_1 \subseteq \cdots$; oracles SEARCH and LABEL; learning parameters $\epsilon, \delta \in (0, 1)$
  1: **initialize** $S_0 \leftarrow \emptyset$, $k_0 \leftarrow 0$.
  2: Draw $x_{1,1}, x_{1,2}$ at random from $D_{\mathcal{X}}$, $T_1 \leftarrow \left\{ (x_{1,1}, \text{LABEL}(x_{1,1})), (x_{1,2}, \text{LABEL}(x_{1,2})) \right\}$
  3: **for** iteration $i = 1, 2, \ldots$ **do**
  4:    $S \leftarrow S_{i-1}$, $k \leftarrow \min \left\{ k' \geq k_{i-1} : H_{k'}(S_{i-1} \cup T_i) \neq \emptyset \right\}$   # Verification stage (Steps 4–13)
  5:    **loop**
  6:      $e \leftarrow \text{SEARCH}_{H_k}(H_k(S \cup T_i))$
  7:      **if** $e \neq \bot$ **then**
  8:        $S \leftarrow S \cup \{e\}$
  9:        $k \leftarrow \min \left\{ k' > k : H_{k'}(S \cup T_i) \neq \emptyset \right\}$
10:      **else**
11:        **break**
12:      **end if**
13:    **end loop**
14:    $S_i \leftarrow S$, $k_i \leftarrow k$                                  # Sampling stage (Steps 14–19)
15:    Define new version space $V_i^{k_i} = H_{k_i}(S_i \cup T_i)$
16:    $T_{i+1} \leftarrow \emptyset$
17:    **for** $j = 1, 2, \ldots, 2^{i+1}$ **do**
18:      $T_{i+1} \leftarrow \text{SAMPLE-AND-LABEL}(V_i^{k_i}, \text{LABEL}, T_{i+1})$
19:    **end for**
20:    **if** $\sigma_{k_i}(2^i, \delta_{i,k_i}) \leq \epsilon$ **then**
21:      **return** any $\hat{h} \in V_i^{k_i}$
22:    **end if**
23: **end for**

---

## C   An Improved Algorithm for the Realizable Case

In this section, we present an improved algorithm for using SEARCH and LABEL in the realizable section. We call this algorithm SEABEL (Algorithm 4).

### C.1   Description of SEABEL

SEABEL proceeds in iterations like LARCH, but takes more advantage of SEARCH. Each iteration is split into two stages: the verification stage, and the sampling stage.

In the verification stage (Steps 4–13), SEABEL makes repeated calls to SEARCH to advance to as high of a complexity class as possible, until $\bot$ is returned. When $\bot$ is returned, it guarantees that whenever the latest version space completely agrees on an unlabeled point, then it is also in agreement with $h^*$, even if it does not contain $h^*$.

In the sampling stage (Steps 14–19), SEABEL performs selective sampling, querying and infering labels based on disagreement over the new version space $V_i^{k_i}$. The preceding verification stage ensures that whenever a label is inferred, it is guaranteed to be in agreement with $h^*$.

The algorithm calls Algorithm 6 in Appendix E, where we slightly abuse the notation in SAMPLE-AND-LABEL that if the counter parameter is missing then it simply does not get updated.

### C.2   Proof of Theorem 2

Observe that $T_{i+1}$ is an iid sample of size $2^{i+1}$ from a distribution (which we call $D_i$) over labeled examples $(x, y)$, where $x \sim D_{\mathcal{X}}$, and

$$y := \begin{cases} V_i^{k_i}(x) & \text{if } x \notin \text{Dis}(V_i^{k_i}), \\ h^*(x) & \text{if } x \in \text{Dis}(V_i^{k_i}), \end{cases}$$

for every $x$ in the support of $D_{\mathcal{X}}$. ($T_1$ is an iid sample from $D_0 := D$; also note $k_0 = 0$ and $S_0 = \emptyset$.)

**Lemma 6.** *Algorithm 4 maintains the following invariants:*

1. *The loop in the verification stage of iteration $i$ terminates for all $i \geq 1$.*

2. *$k_i \leq k^*$ for all $i \geq 0$.*

3. *$h^*(x) = V_i^{k_i}(x)$ for all $x \notin \text{Dis}(V_i^{k_i})$ for all $i \geq 1$.*

4. *$h^*$ is consistent with $S_i \cup T_{i+1}$ for all $i \geq 0$.*

*Proof.* It is easy to see that $S$ only contains examples provided by SEARCH, and hence the labels are consistent with $h^*$.

Now we prove that the invariants hold by induction on $i$, starting with $i = 0$. For the base case, only the last invariant needs checking, and it is true because the labels in $T_1$ are obtained from LABEL.

For the inductive step, fix any $i \geq 1$, and assume that $k_{i-1} \leq k^*$, and that $h^*$ is consistent with $T_i$. Now consider the verification stage in iteration $i$. We first prove that the loop in the verification stage will terminate and establish some properties upon termination. Observe that $k$ and $S$ are initially $k_{i-1}$ and $S_{i-1}$, respectively. Throughout the loop, the examples added to $S$ are obtained from SEARCH, and hence are consistent with $h^*$. Thus, $h^* \in H_{k^*}(S \cup T_i)$, implying $H_{k^*}(S \cup T_i) \neq \emptyset$. If $k = k^*$, then $\text{SEARCH}_{H_{k^*}}(H_{k^*}(S \cup T_i))$ would return $\perp$ and Algorithm 4 would exit the loop. If $\text{SEARCH}_{H_k}(H_k(S \cup T_i)) \neq \perp$, then $k < k^*$, and $k$ cannot be increased beyond $k^*$ since $H_{k^*}(S \cup T_i) \neq \emptyset$. Thus, the loop must terminate with $k \leq k^*$, implying $k_i \leq k^*$. This establishes invariants 1 and 2. Moreover, because the loop terminates with $\text{SEARCH}_{H_k}(H_k(S \cup T_i))$ returning $\perp$ (and here, $k = k_i$ and $H_k(S \cup T_i) = V_i^{k_i}$), there is no counterexample $x \in \mathcal{X}$ such that $h^*$ disagrees with every $h \in V_i^{k_i}$. This implies that $h^*(x) = V_i^{k_i}(x)$ for all $x \notin \text{Dis}(V_i^{k_i})$, i.e., invariant 3.

Now consider any $(x, y)$ added to $T_{i+1}$ in the sampling stage. If $x \in \text{Dis}(V_i^{k_i})$, the label is obtained from LABEL, and hence is consistent with $h^*$; if $x \notin \text{Dis}(V_i^{k_i})$, the label is $V_i^{k_i}(x)$, which is the same as $h^*(x)$ as previously argued. So $h^*$ is consistent with all examples in $T_{i+1}$, and hence also all examples in $S_i \cup T_{i+1}$, proving invariant 4. This completes the induction. $\square$

Let $E_i$ be the event in which the following hold:

1. For every $k \geq 0$, every $h \in H_k$ satisfies

$$\text{err}(h, D_{i-1}) \leq \text{err}(h, T_i) + \sqrt{\text{err}(h, T_i)\sigma_k(2^i, \delta_{i,k})} + \sigma_k(2^i, \delta_{i,k}).$$

2. The number of LABEL queries in iteration $i$ (to form $T_{i+1}$) is at most

$$2^{i+1} \Pr_{x \sim D_{\mathcal{X}}}[x \in \text{Dis}(V_i^{k_i})] + O\left(\sqrt{2^{i+1} \Pr_{x \sim D_{\mathcal{X}}}[x \in \text{Dis}(V_i^{k_i})] \log(1/\delta_i)} + \log(1/\delta_i)\right),$$

Using Theorem 5 and Lemma 3, along with the union bound, $\Pr(E_i) \geq 1 - \delta_i$. Define $E := \cap_{i=1}^{\infty} E_i$; a union bound implies that $\Pr(E) \geq 1 - \delta$.

We now prove Theorem 2, starting with the error rate guarantee. Condition on the event $E$. Since $k_i \leq k^*$ (Lemma 6), the definition of $\sigma_k$ from (3), the halting condition in Algorithm 4, and Fact 1 imply that the algorithm must halt after at most $I$ iterations, where

$$2^I \leq O\left(\frac{1}{\epsilon}\left(d_{k^*} \log \frac{1}{\epsilon} + \log \frac{k^*}{\delta}\right)\right). \tag{6}$$

So let $I$ denote the iteration in which Algorithm 4 halts. By definition of $E_I$, we have

$$\text{err}(\hat{h}, D_{i-1}) \leq \text{err}(\hat{h}, T_i) + \sqrt{\text{err}(\hat{h}, T_i)\sigma_{k_i}(2^i, \delta_{i,k_i})} + \sigma_{k_i}(2^i, \delta_{i,k_i})$$
$$= \sigma_{k_i}(2^i, \delta_{i,k_i}) \leq \epsilon,$$

where the second inequality follows from the termination condition. By Lemma 6, $h^*(x) = V_{i-1}^{k_{i-1}-1}(x)$ for all $x \notin \text{Dis}(V_{i-1}^{k_{i-1}-1})$. Therefore, $D(\cdot \mid x) = D_{i-1}(\cdot \mid x)$ for every $x$ in the support of $D_{\mathcal{X}}$, and

$$\text{err}(\hat{h}, D) = \text{err}(\hat{h}, D_{i-1}) \leq \epsilon.$$

Now we bound the unlabeled, LABEL, and SEARCH complexities, all conditioned on event $E$. First, as argued above, the algorithm halts after at most $I$ iterations, where $2^I$ is bounded as in (6). The number of unlabeled examples drawn from $D_{\mathcal{X}}$ across all iterations is within a factor of two of the number of examples drawn in the final sampling stage, which is $O(2^I)$. Thus (6) also gives the bound on the number of unlabeled examples drawn.

Next, we consider the SEARCH complexity. For each iteration $i$, each call to SEARCH either returns a counterexample that forces $k$ to increment (but never past $k^*$, as implied by Lemma 6), or returns $\perp$ which causes an exit from the verification stage loop. Therefore, the total number of SEARCH calls is at most

$$k^* + I = k^* + O\left(\log \frac{d_{k^*}}{\epsilon} + \log \log \frac{k^*}{\delta}\right).$$

Finally, we consider the LABEL complexity. For $i \leq I$, we first show that the version space $V_i^{k_i}$ is always contained in a ball of small radius (with respect to the disagreement pseudometric). Specifically, for every $h, h'$ in $V_i^{k_i}$, $\text{err}(h, T_i) = 0$ and $\text{err}(h, T_i) = 0$. By definition of $E_i$, this implies that

$$\text{err}(h, D_{i-1}) \leq \sigma_{k_i}(2^i, \delta_{i,k_i}) \quad \text{and} \quad \text{err}(h', D_{i-1}) \leq \sigma_{k_i}(2^i, \delta_{i,k_i}).$$

Therefore, by the triangle inequality and the fact $k_i \leq k^*$,

$$\Pr_{x \sim D}[h(x) \neq h'(x)] \leq 2\sigma_{k_i}(2^i, \delta_{i,k_i}) \leq 2\sigma_{k^*}(2^i, \delta_{i,k^*}).$$

Also, the upper bound $2^I \leq \tilde{O}(d_{k^*}/\epsilon)$ from (6) implies the lower bound $\sigma_{k^*}(2^i, \delta_{i,k^*}) \geq \epsilon/2$ for $i \leq I$. Thus, the probability mass of the disagreement region can be bounded as

$$\Pr_{x \sim D_{\mathcal{X}}}[x \in \text{Dis}(V_i^{k_i})] \leq \theta_{k_i}(\epsilon) \cdot 2\sigma_{k^*}(2^i, \delta_{i,k^*}).$$

By definition of $E_i$, the number of queries to LABEL at iteration $i$ is at most

$$2^{i+1} \Pr_{x \sim D_{\mathcal{X}}}[x \in \text{Dis}(V_i^{k_i})] + O\left(\sqrt{2^{i+1} \Pr_{x \sim D_{\mathcal{X}}}[x \in \text{Dis}(V_i^{k_i})] \log(1/\delta_{i,k})} + \log(1/\delta_i)\right),$$

which is at most

$$O\left(2^i \cdot \theta_{k_i}(\epsilon) \cdot \sigma_{k^*}(2^i, \delta_{i,k^*})\right).$$

We conclude that the total number of LABEL queries by Algorithm 4 is bounded by

$$2 + \sum_{i=1}^{I} O\left(2^i \cdot \theta_{k_i}(\epsilon) \cdot \sigma_{k^*}(2^i, \delta_{i,k^*})\right)$$

$$= 2 + \sum_{i=1}^{I} O\left(2^i \cdot \max_{k \leq k^*} \theta_k(\epsilon) \cdot \sigma_{k^*}(2^i, \delta_{i,k^*})\right)$$

$$= O\left(\max_{k \leq k^*} \theta_k(\epsilon) \cdot \left(\sum_{i=1}^{I} 2^i \cdot \frac{d \ln(2^i) + \ln(\frac{(i^2+i)(k^*)^2}{\delta})}{2^i}\right)\right)$$

$$= O\left(\max_{k \leq k^*} \theta_k(\epsilon) \cdot \left(d_{k^*} I^2 + I \log \frac{k^*}{\delta}\right)\right)$$

$$= O\left(\max_{k \leq k^*} \theta_k(\epsilon) \cdot \left(d_{k^*} \left(\log \frac{d_{k^*}}{\epsilon} + \log \log \frac{k^*}{\delta}\right)^2 + \left(\log \frac{d_{k^*}}{\epsilon} + \log \log \frac{k^*}{\delta}\right) \log \frac{k^*}{\delta}\right)\right)$$

$$= \tilde{O}\left(\max_{k \leq k^*} \theta_k(\epsilon) \cdot \left(d_{k^*} \cdot \log^2 \frac{1}{\epsilon} + \log k^*\right)\right)$$

as claimed. $\qquad \square$

# D A-LARCH: An Adaptive Agnostic Algorithm

In this section, we present a generalization of SEABEL that works in the agnostic setting. We call this algorithm A-LARCH (Algorithm 5).

## D.1 Description of A-LARCH

A-LARCH proceeds in iterations like SEABEL. Each iteration is split into three stages: the error estimation stage, the verification stage, and the sampling stage.

In the error estimation stage, A-LARCH uses a structural risk minimization approach (Step 4) to compute $\gamma_{i-1}$, a (tight) upper bound on $\Pr[h^*(x) \neq y, x \in \mathrm{Dis}(V_{i-1})]$. (See item 1 of Lemma 7 for justification.)

The verification stage (Steps 5–18) and sampling stage (Steps 20–23) in A-LARCH are similar to the corresponding stages in SEABEL.

Same as SEABEL, the algorithm calls Algorithm 6, 8 and 9 in Appendix E (SAMPLE-AND-LABEL, PRUNE-VERSION-SPACE and UPGRADE-VERSION-SPACE, respectively), where we slightly abuse the notation in SAMPLE-AND-LABEL that if the counter parameter is missing then it simply does not get updated.

---

**Algorithm 5** A-LARCH

---

**input:** Nested hypothesis set $H_0 \subseteq H_1 \subseteq \cdots$; oracles LABEL and SEARCH; learning parameter $\delta \in (0,1)$; unlabeled examples budget $m = 2^{I+2}$.

**output:** hypothesis $\hat{h}$.

1: **initialize** $S \leftarrow \emptyset$, $k_0 \leftarrow 0$.
2: Draw $x_{1,1}, x_{1,2}$ at random from $D_{\mathcal{X}}$, $T_1 \leftarrow \big\{(x_{1,1}, \mathrm{LABEL}(x_{1,1})), (x_{1,2}, \mathrm{LABEL}(x_{1,2}))\big\}$
3: **for** $i = 1, 2, \ldots, I$ **do**
4:      $\gamma_{i-1} \leftarrow \min_{k' \geq k_{i-1}, h \in H_{k'}} \left\{ \mathrm{err}(h, T_i) + \sqrt{\mathrm{err}(h, T_i)\sigma_{k'}(2^i, \delta_{i,k'})} + \sigma_{k'}(2^i, \delta_{i,k'}) \right\}$ # Error estimation stage (Step 4)
5:      $S \leftarrow S_{i-1}$, $k \leftarrow k_{i-1}$.                # Verification stage (Steps 5–18)
6:      $V_i^k \leftarrow$ PRUNE-VERSION-SPACE$(H_k(S), T_i, \delta_i)$
7:      **loop**
8:          **if** $\min_{h \in H_k(S)} \mathrm{err}(h, T_i) > \gamma_{i-1} + \sqrt{\gamma_{i-1}\sigma_k(2^i, \delta_{i,k})} + \sigma_k(2^i, \delta_{i,k})$ **then**
9:              $(k, S, V_i^k) \leftarrow$ UPGRADE-VERSION-SPACE$(k, S, \emptyset)$
10:          **else**
11:              $e \leftarrow$ SEARCH$_{H_k}(V_i^k)$
12:              **if** $e \neq \perp$ **then**
13:                  $(k, S, V_i^k) \leftarrow$ UPGRADE-VERSION-SPACE$(k, S, \{e\})$
14:              **else**
15:                  **break**
16:              **end if**
17:          **end if**
18:      **end loop**
19:      $S_i \leftarrow S$, $k_i \leftarrow k$
20:      $T_{i+1} \leftarrow \emptyset$                         # Sampling stage (Steps 20–23)
21:      **for** $j = 1, 2, \ldots, 2^{i+1}$ **do**
22:          $T_{i+1} \leftarrow$ SAMPLE-AND-LABEL$(V_i^{k_i}, \mathrm{LABEL}, T_{i+1})$
23:      **end for**
24: **end for**
25: **return** any $\hat{h} \in V_I^{k_I}$.

---

## D.2 Proof of Theorem 3

Let

$$
\begin{aligned}
M(\nu, k^*, \epsilon, \delta) \quad &:= \quad \min\left\{2^n : n \in \mathbb{N},\ 6\sqrt{\nu\sigma_{k^*}(2^n, \delta_{n,k^*})} + 21\sigma_{k^*}(2^n, \delta_{n,k^*}) \le \epsilon\right\} \\
&= \quad O\left(\frac{(d_{k^*}\log(1/\epsilon) + \log(k^*/\delta))(\nu + \epsilon)}{\epsilon^2}\right).
\end{aligned}
$$

where the second line is from Fact 1.

**Theorem 6** (Restatement of Theorem 3). *Assume* $\mathrm{err}(h^*) = \nu$. *If Algorithm 5 is run with inputs hypothesis classes* $\{H_k\}_{k=0}^{\infty}$*, oracles* SEARCH *and* LABEL*, learning parameter* $\delta$*, unlabeled examples budget* $m = M(\nu, k^*, \epsilon, \delta)$ *and the disagreement coefficient of* $H_k(S)$ *is at most* $\theta_k(\cdot)$*, then, with probability* $1 - \delta$:
*(1) The returned hypothesis* $\hat{h}$ *satisfies*

$$
\mathrm{err}(\hat{h}) \le \nu + \epsilon\,.
$$

*(2) The total number of queries to oracle* SEARCH *is at most*

$$
k^* + \log m \le k^* + O\left(\log\frac{d_{k^*}}{\epsilon} + \log\log\frac{k^*}{\delta}\right).
$$

*(3) The total number of queries to oracle* LABEL *is at most*

$$
\tilde{O}\left(\max_{k \le k^*}\theta_k(2\nu + 2\epsilon) \cdot d_{k^*}\left(\log\frac{1}{\epsilon}\right)^2 \cdot \left(1 + \frac{\nu^2}{\epsilon^2}\right)\right).
$$

The proof relies on an auxillary lemma. First, we need to introduce the following notation.

Observe that $T_{i+1}$ is an iid sample of size $2^{i+1}$ from a distribution (which we call $D_i$) over labeled examples $(x, y)$, where $x \sim D_{\mathcal{X}}$ and the conditional distribution is

$$
D_i(y \mid x) := \begin{cases} \mathbb{1}\{y = V_i^{k_i}(x)\} & \text{if } x \notin \mathrm{Dis}(V_i^{k_i})\,, \\ D(y \mid x) & \text{if } x \in \mathrm{Dis}(V_i^{k_i})\,. \end{cases}
$$

$T_1$ is a sample of size 2 from $D_0 := D$.

Let $E_i$ be the event in which the following hold:

1. For every $k \ge 0$, every $h \in H_k$ satisfies

$$
\begin{aligned}
\mathrm{err}(h, D_{i-1}) &\le \mathrm{err}(h, T_i) + \sqrt{\mathrm{err}(h, T_i)\sigma_k(2^i, \delta_{i,k})} + \sigma_k(2^i, \delta_{i,k})\,, \\
\mathrm{err}(h, T_i) &\le \mathrm{err}(h, D_{i-1}) + \sqrt{\mathrm{err}(h, D_{i-1})\sigma_k(2^i, \delta_{i,k})} + \sigma_k(2^i, \delta_{i,k})\,.
\end{aligned}
$$

2. The number of LABEL queries at iteration $i$ is at most

$$
2^{i+1}\Pr_{x \sim D_{\mathcal{X}}}[x \in \mathrm{Dis}(V_i^{k_i})] + O\left(\sqrt{2^{i+1}\Pr_{x \sim D_{\mathcal{X}}}[x \in \mathrm{Dis}(V_i^{k_i})]\log(1/\delta_i)} + \log(1/\delta_i)\right).
$$

Using Theorem 5 and Lemma 3, along with the union bound, $\Pr(E_i) \ge 1 - \delta_i$. Define $E := \cap_{i=1}^{\infty} E_i$, by union bound, $\Pr(E) \ge 1 - \delta$. Recall that $k_i$ is the value of $k$ at the end of iteration $i$.

**Lemma 7.** *On event E, Algorithm 5 maintains the following invariants:*

1. *For all* $i \ge 1$, $\gamma_{i-1}$ *is such that*

$$
\mathrm{err}(h^*, D_{i-1}) \le \gamma_{i-1} \le \mathrm{err}(h^*, D_{i-1}) + 2\sqrt{\mathrm{err}(h^*, D_{i-1})\sigma_{k^*}(2^i, \delta_{i,k^*})} + 3\sigma_{k^*}(2^i, \delta_{i,k^*}).
$$

2. *The loop in the verification stage of iteration* $i$ *terminates for all* $i \ge 1$.

3. $k_i \leq k^*$ *for all* $i \geq 0$.

4. $h^*(x) = V_i^{k_i}(x)$ *for all* $x \notin \mathrm{Dis}(V_i^{k_i})$ *for all* $i \geq 1$.

5. *For all* $i \geq 0$, *for every hypothesis* $h$, $\mathrm{err}(h, D_i) - \mathrm{err}(h^*, D_i) \geq \mathrm{err}(h, D) - \mathrm{err}(h^*, D)$. *Therefore,* $h^*$ *is the optimal hypothesis among* $\cup_k H_k$ *with respect to* $D_i$.

*Proof.* Throughout, we assume the event $E$ holds.

It is easy to see that $S$ only contains examples provided by SEARCH, and hence the labels are consistent with $h^*$.

Now we prove that the invariants hold by induction on $i$, starting with $i = 0$. For the base case, invariant 3 holds since $k_0 = 0 \leq k^*$, and invariant 5 holds since $D_0 = D$ and $h^*$ is the optimal hypothesis in $\cup_k H_k$.

Now consider the inductive step. We first prove that invariant 1 holds.

(1) By definition of $E_i$, for all $k' \geq k_{i-1}$, we have for all $h \in H_{k'}$,

$$\mathrm{err}(h, D_{i-1}) \leq \mathrm{err}(h, T_i) + \sqrt{\mathrm{err}(h, T_i)\sigma_{k'}(2^i, \delta_{i,k'})} + \sigma_{k'}(2^i, \delta_{i,k'}).$$

Thus,

$$\min_{h \in H_{k'}} \mathrm{err}(h, D_{i-1}) \leq \min_{h \in H_{k'}} \mathrm{err}(h, T_i) + \sqrt{\mathrm{err}(h, T_i)\sigma_{k'}(2^i, \delta_{i,k'})} + \sigma_{k'}(2^i, \delta_{i,k'}).$$

Taking minimum over $k' \geq k_{i-1}$ on both sides, notice that $h^*$ is the optimal hypothesis with respect to $D_{i-1}$ and recall the definition of $\gamma_{i-1}$, we get

$$\mathrm{err}(h^*, D_{i-1}) \leq \gamma_{i-1}.$$

(2) By definition of $\gamma_{i-1}$, we have

$$\gamma_{i-1} = \min_{k' \geq k_{i-1}, h \in H_{k'}} \left\{ \mathrm{err}(h, T_i) + \sqrt{\mathrm{err}(h, T_i)\sigma_{k'}(2^i, \delta_{i,k'})} + \sigma_{k'}(2^i, \delta_{i,k'}) \right\}$$

Taking $k' = k^*$, $h = h^*$, we get

$$\gamma_{i-1} \leq \mathrm{err}(h^*, T_i) + \sqrt{\mathrm{err}(h^*, T_i)\sigma_{k^*}(2^i, \delta_{i,k^*})} + \sigma_{k^*}(2^i, \delta_{i,k^*})$$

In conjunction with the fact that by definition of $E_i$,

$$\mathrm{err}(h^*, T_i) \leq \mathrm{err}(h^*, D_{i-1}) + \sqrt{\mathrm{err}(h^*, D_{i-1})\sigma_{k^*}(2^i, \delta_{i,k^*})} + \sigma_{k^*}(2^i, \delta_{i,k^*})$$

We get

$$\gamma_{i-1} \leq \mathrm{err}(h^*, D_{i-1}) + 2\sqrt{\mathrm{err}(h^*, D_{i-1})\sigma_{k^*}(2^i, \delta_{i,k^*})} + 3\sigma_{k^*}(2^i, \delta_{i,k^*}).$$

Thus, invariant 1 is established for iteration $i$.

Now consider the verification stage in iteration $i$. We first prove that the loop in the verification stage will terminate and establish some properties upon termination. Observe that $k$ and $S$ are initially $k_{i-1}$ and $S_{i-1}$, respectively. Throughout the loop, the examples added to $S$ are obtained from SEARCH, and hence are consistent with $h^*$. In addition, we have the following claim regarding $k^*$.

**Claim 2.** *If invariants 1–5 holds for iteration* $i - 1$, *then for iteration* $i$, *the following holds:*

*(a)* $\min_{h \in H_{k^*}(S)} \mathrm{err}(h, T_i) \leq \gamma_{i-1} + \sqrt{\gamma_{i-1}\sigma_{k^*}(2^i, \delta_{i,k^*})} + \sigma_{k^*}(2^i, \delta_{i,k^*})$

*(b)*

$$h^* \in V_i^{k^*} = \Big\{ h \in H_{k^*}(S) :$$
$$\mathrm{err}(h, T_i) \leq \min_{h' \in H_{k^*}(S)} \mathrm{err}(h', T_i) + 2\sqrt{\mathrm{err}(h', T_i)\sigma_{k^*}(2^i, \delta_{i,k^*})} + 3\sigma_{k^*}(2^i, \delta_{i,k^*}) \Big\}.$$

*Proof.* Recall that $h^*$ is the optimal hypothesis under distribution $D_{i-1}$. We have already shown above that $\mathrm{err}(h^*, D_{i-1}) \leq \gamma_{i-1}$. By the definition of $E_i$,

$$
\begin{aligned}
\min_{h \in H_{k^*}(S)} \mathrm{err}(h, T_i) \quad &\leq \quad \mathrm{err}(h^*, T_i) \\
&\leq \quad \mathrm{err}(h^*, D_{i-1}) + \sqrt{\mathrm{err}(h^*, D_{i-1})\sigma(2^i, \delta_{i,k^*})} + \sigma(2^i, \delta_{i,k^*}) \\
&\leq \quad \gamma_{i-1} + \sqrt{\gamma_{i-1}\sigma_{k^*}(2^i, \delta_{i,k^*})} + \sigma_{k^*}(2^i, \delta_{i,k^*})
\end{aligned}
$$

where the last inequality is from that $\mathrm{err}(h^*, D_{i-1}) \leq \gamma_{i-1}$. This proves item (a).

On the other hand, for all $h'$ in $H_{k^*}(S)$,

$$
\begin{aligned}
\mathrm{err}(h^*, T_i) \quad &\leq \quad \mathrm{err}(h^*, D_{i-1}) + \sqrt{\mathrm{err}(h^*, D_{i-1})\sigma_{k^*}(2^i, \delta_{i,k^*})} + \sigma(2^i, \delta_{i,k^*}) \\
&\leq \quad \mathrm{err}(h', D_{i-1}) + \sqrt{\mathrm{err}(h', D_{i-1})\sigma_{k^*}(2^i, \delta_{i,k^*})} + \sigma(2^i, \delta_{i,k^*}) \\
&\leq \quad \mathrm{err}(h', T_i) + 2\sqrt{\mathrm{err}(h', T_i)\sigma_{k^*}(2^i, \delta_{i,k^*})} + 3\sigma(2^i, \delta_{i,k^*}).
\end{aligned}
$$

where the first inequality is from the definition of $E_i$, the second inequality is from Invariant 5 of iteration $i - 1$, the third inequality is from the definition of $E_i$.

Thus, $\mathrm{err}(h^*, T_i) \leq \min_{h' \in H_{k^*}(S)} \mathrm{err}(h', T_i) + 2\sqrt{\mathrm{err}(h', T_i)\sigma_{k^*}(2^i, \delta_{i,k^*})} + 3\sigma_{k^*}(2^i, \delta_{i,k^*})$, proving item (b). $\qquad\square$

Claim 2 implies that $k$ cannot increase beyond $k^*$. To see this, observe that Claim 2(a) implies the condition in Step 8 is not satisfied for $k = k^*$. In addition, Claim 2(b) implies that $h^* \in V_i^{k^*} \neq \emptyset$, which in turn means that $\mathrm{SEARCH}_{H_{k^*}}(V_i^{k^*}) = \bot$. Hence, the loop in the verification stage would terminate if $k$ ever reaches $k^*$. Because iteration $i$ starts with $k \leq k^*$ (as invariant 3 holds in iteration $i - 1$), invariants 2 and 3 must also hold for iteration $i$.

Finally, we can establish invariants 4 and 5 for iteration $i$. Because the loop terminates with $\mathrm{SEARCH}_{H_k}(V_i^{k_i})$ returning $\bot$, there is no counterexample $x \in \mathcal{X}$ such that $h^*$ disagrees with every $h \in V_i^{k_i}$. This implies that $h^*(x) = V_i^{k_i}(x)$ for all $x \notin \mathrm{Dis}(V_i^{k_i})$ (i.e., invariant 4). Hence, for any hypothesis $h$,

$$
\mathrm{err}(h, D_i) = \Pr[h(x) \neq h^*(x), x \notin \mathrm{Dis}(V_i^{k_i})] + \Pr[h(x) \neq y, x \in \mathrm{Dis}(V_i^{k_i})].
$$

Therefore,

$$
\begin{aligned}
&\mathrm{err}(h, D_i) - \mathrm{err}(h^*, D_i) \\
&= \Pr[h(x) \neq h^*(x), x \notin \mathrm{Dis}(V_i^{k_i})] + \Pr[h(x) \neq y, x \in \mathrm{Dis}(V_i^{k_i})] \\
&\quad - \Pr[h^*(x) \neq y, x \in \mathrm{Dis}(V_i^{k_i})] \\
&\geq \Pr[h(x) \neq y, x \notin \mathrm{Dis}(V_i^{k_i})] - \Pr[h^*(x) \neq y, x \notin \mathrm{Dis}(V_i^{k_i})] \\
&\quad + \Pr[h(x) \neq y, x \in \mathrm{Dis}(V_i^{k_i})] - \Pr[h^*(x) \neq y, x \in \mathrm{Dis}(V_i^{k_i})] \\
&= \mathrm{err}(h, D) - \mathrm{err}(h^*, D),
\end{aligned}
$$

which proves invariant 5 for iteration $i$. $\qquad\square$

*Proof of Theorem 6.* Supose event $E$ happens.

We first show a claim regarding the error of hypotheses in current version spaces.

**Claim 3.** *On event $E$, for all $i \geq 1$, for all $h \in V_i^{k_i}$,*

$$
\mathrm{err}(h, D) \leq \mathrm{err}(h^*, D_{i-1}) + 6\sqrt{\mathrm{err}(h^*, D_{i-1})\sigma_{k^*}(2^i, \delta_{i,k^*})} + 21\sigma_{k^*}(2^i, \delta_{i,k^*}).
$$

*Proof.* First, for every $h$ in $V_i^{k_i}$,

$$\text{err}(h, T_i) \le \min_{h' \in H_{k_i}(S)} \text{err}(h', T_i) + 2\sqrt{\text{err}(h', T_i)\sigma_{k_i}(2^i, \delta_{i,k_i})} + 3\sigma_{k_i}(2^i, \delta_{i,k_i}),$$

and since the condition in step 8 is not satisfied for $k = k_i$, we know that

$$\min_{h' \in H_{k_i}(S)} \text{err}(h', T_i) \le \gamma_{i-1} + \sqrt{\gamma_{i-1}\sigma_{k_i}(2^i, \delta_{i,k_i})} + \sigma_{k_i}(2^i, \delta_{i,k_i}).$$

Thus,

$$\text{err}(h, T_i) \le \gamma_{i-1} + 3\sqrt{\gamma_{i-1}\sigma_{k_i}(2^i, \delta_{i,k_i})} + 6\sigma_{k_i}(2^i, \delta_{i,k_i}). \tag{7}$$

By definition of event $E_i$, we also have

$$\text{err}(h, D_{i-1}) \le \text{err}(h, T_i) + \sqrt{\text{err}(h, T_i)\sigma_{k_i}(2^i, \delta_{i,k_i})} + \sigma_{k_i}(2^i, \delta_{i,k_i}).$$

Hence,

$$\text{err}(h, D_{i-1}) \le \gamma_{i-1} + 4\sqrt{\gamma_{i-1}\sigma_{k_i}(2^i, \delta_{i,k_i})} + 10\sigma_{k_i}(2^i, \delta_{i,k_i}).$$

Furthermore, by item 1 of Lemma 7,

$$\gamma_{i-1} \le \text{err}(h^*, D_{i-1}) + 2\sqrt{\text{err}(h^*, D_{i-1})\sigma_{k^*}(2^i, \delta_{i,k^*})} + 3\sigma_{k^*}(2^i, \delta_{i,k^*}).$$

This implies that

$$
\begin{aligned}
\text{err}(h, D_{i-1}) &\le \text{err}(h^*, D_{i-1}) + 6\sqrt{\text{err}(h^*, D_{i-1})\sigma_{k_i}(2^i, \delta_{i,k_i})} + 21\sigma_{k_i}(2^i, \delta_{i,k_i}) \\
&\le \text{err}(h^*, D_{i-1}) + 6\sqrt{\text{err}(h^*, D_{i-1})\sigma_{k^*}(2^i, \delta_{i,k^*})} + 21\sigma_{k^*}(2^i, \delta_{i,k^*}). \quad \square
\end{aligned}
$$

where the second inequality is from item 3 of Lemma 7.

We first prove the error rate guarantee. Suppose iteration $i = I = \log_2 M(\nu, k^*, \epsilon, \delta)$ has been reached. Observe that from Claim 3, for $\hat{h} \in V_I^{k_I}$,

$$\text{err}(\hat{h}, D_{I-1}) - \text{err}(h^*, D_{I-1}) \le 6\sqrt{\text{err}(h^*, D_{I-1})\sigma_{k^*}(2^I, \delta_{I,k^*})} + 21\sigma_{k^*}(2^I, \delta_{I,k^*}) \le \epsilon$$

where the second inequality is from that $m = 2^I = M(\nu, k^*, \epsilon, \delta)$. Thus, by item 5 of Lemma 7,

$$\text{err}(\hat{h}, D) - \text{err}(h^*, D) \le \text{err}(\hat{h}, D_{I-1}) - \text{err}(h^*, D_{I-1}) \le \epsilon.$$

Next, we prove the bound on the number of SEARCH queries. From Lemma 7, Algorithm 5 maintains the invariant that $k \le k^*$. For each iteration $i$, each call to SEARCH either returns an example forcing $k$ to increment, or returns $\perp$ which causes an exit from the verification stage loop. Therefore, the total number of SEARCH calls is at most

$$k^* + I \le k^* + O\left(\log \frac{d_{k^*}}{\epsilon^2} + \log\log \frac{k^*}{\delta}\right).$$

Finally, we prove the bound on the number of LABEL queries. This is done in a few steps.

1. We first show that the version space $V_i^{k_i}$ is always contained in a ball of small radius (with respect to the disagreement pseudometric $\Pr_{x \sim D_\mathcal{X}}[h(x) \ne h'(x)]$). Specifically, for $1 \le i \le I$, for any $h, h' \in V_i^{k_i}$, from Claim 3, in conjunction with triangle inequality, and that $\text{err}(h^*, D_{i-1}) \le \text{err}(h, D) = \nu$,

$$
\begin{aligned}
&\Pr_{x \sim D_\mathcal{X}}[h(x) \ne h'(x)] \\
&\le 2\,\text{err}(h^*, D_{i-1}) + 12\sqrt{\text{err}(h^*, D_{i-1})\sigma_{k^*}(2^i, \delta_{i,k^*})} + 42\sigma_{k^*}(2^i, \delta_{i,k^*}) \\
&\le 2\nu + 12\sqrt{\nu\sigma_{k^*}(2^i, \delta_{i,k^*})} + 42\sigma_{k^*}(2^i, \delta_{i,k^*})
\end{aligned}
$$

Thus, $V_i^{k_i}$ is contained in $B_{H_{k_i}(S)}(h, 2\nu + 12\sqrt{\nu\sigma_{k^*}(2^i, \delta_{i,k^*})} + 42\sigma_{k^*}(2^i, \delta_{i,k^*}))$ for some $h$ in $H_{k_i}(S)$.

2. Next we bound the label complexity per iteration. Note that by the choice of $m = 2^I = M(\nu, k^*, \epsilon, \delta)$, $6\sqrt{\nu \sigma_{k^*}(2^{I-1}, \delta_{I-1,k^*})} + 21\sigma_{k^*}(2^{I-1}, \delta_{I-1,k^*}) \geq \epsilon$, therefore for all $1 \leq i \leq I$, $6\sqrt{\nu \sigma_{k^*}(2^i, \delta_{i,k^*})} + 21\sigma_{k^*}(2^i, \delta_{i,k^*}) \geq \epsilon/2$. Thus, the size of the disagreement region can be bounded as

$$
\begin{aligned}
\Pr_{x \sim D_{\mathcal{X}}}[x \in \mathrm{Dis}(V_i^{k_i})] &\leq \theta_k(2\nu + \epsilon) \cdot \left(2\nu + 12\sqrt{\nu \sigma_{k^*}(2^i, \delta_{i,k^*})} + 42\sigma_{k^*}(2^i, \delta_{i,k^*})\right) \\
&\leq \theta_k(2\nu + \epsilon) \cdot \left(8\nu + 48\sigma_{k^*}(2^i, \delta_{i,k^*})\right).
\end{aligned}
\tag{8}
$$

By definition of $E_i$, the number of queries to LABEL at iteration $i$ is at most

$$
2^{i+1} \Pr_{x \sim D_{\mathcal{X}}}[x \in \mathrm{Dis}(V_i^{k_i})] + O\left(\sqrt{2^{i+1} \Pr_{x \sim D_{\mathcal{X}}}[x \in \mathrm{Dis}(V_i^{k_i})] \log(1/\delta_{i,k^*})} + \log(1/\delta_{i,k^*})\right).
$$

Combining this with (8) gives

$$
\text{\# LABEL queries in iteration } i = O\left(2^i \cdot \theta_{k_i}(2\nu + \epsilon) \cdot (\nu + \sigma_{k^*}(2^i, \delta_{i,k^*}))\right).
\tag{9}
$$

3. From the setting of $m = 2^I = \tilde{O}(d_{k^*}(\nu + \epsilon)/\epsilon^2)$, we get that

$$
I = O\left(\log \frac{d_{k^*}}{\epsilon} + \log\log \frac{k^*}{\delta}\right).
$$

Now, using (9), we get that the total number of LABEL queries by Algorithm 5 is bounded by

$$
2 + \sum_{i=1}^I O\left(2^i \cdot \theta_{k_i}(2\nu + \epsilon) \cdot (\nu + \sigma_{k^*}(2^i, \delta_{i,k^*}))\right)
$$

$$
= 2 + \sum_{i=1}^I O\left(2^i \cdot \max_{k \leq k^*} \theta_k(2\nu + \epsilon) \cdot (\nu + \sigma_{k^*}(2^i, \delta_{i,k^*}))\right)
$$

$$
= O\left(\max_{k \leq k^*} \theta_k(2\nu + \epsilon) \cdot \left(\sum_{i=1}^I 2^i(\nu + \sigma_{k^*}(2^i, \delta_{i,k^*}))\right)\right)
$$

$$
= O\left(\max_{k \leq k^*} \theta_k(2\nu + \epsilon) \cdot \left(\nu 2^I + \sum_{i=1}^I 2^i \frac{d \ln(2^i) + \ln(\frac{(i^2+i)(k^*)^2}{\delta})}{2^i}\right)\right)
$$

$$
= O\left(\max_{k \leq k^*} \theta_k(2\nu + \epsilon) \cdot \left(\nu 2^I + d_{k^*} I^2 + I \log \frac{k^*}{\delta}\right)\right)
$$

$$
= O\left(\max_{k \leq k^*} \theta_k(2\nu + \epsilon) \cdot \left(\frac{\nu^2 + \epsilon \nu}{\epsilon^2}\left(d_{k^*} \log \frac{1}{\epsilon} + \log \frac{k^*}{\delta}\right) + d_{k^*}\left(\log \frac{d_{k^*}}{\epsilon} + \log\log \frac{k^*}{\delta}\right)^2 \right.\right.
$$

$$
\left.\left. + \left(\log \frac{d_{k^*}}{\epsilon} + \log\log \frac{k^*}{\delta}\right)\log \frac{k^*}{\delta}\right)\right)
$$

$$
= \tilde{O}\left(\max_{k \leq k^*} \theta_k(2\nu + \epsilon) \cdot \left(d_{k^*}(\log \frac{1}{\epsilon})^2 + \log \frac{k^*}{\delta}\right) \cdot \left(1 + \frac{\nu^2}{\epsilon^2}\right)\right). \qquad \square
$$

# E  Performance Guarantees of AA-LARCH

## E.1  Detailed Description of Subroutines

Subroutine SAMPLE-AND-LABEL performs standard disagreement-based selective sampling. Specifically, it draws an unlabeled example $x$ from the $D_{\mathcal{X}}$. If $x$ is in the agreement region of version space

$V$, its label is inferred as $V(x)$; otherwise, we query the LABEL oracle to get its label. The counter $c$ is incremented when LABEL is called.

---

**Algorithm 6** SAMPLE-AND-LABEL

---

**input:** Version space $V \subset H$, oracle LABEL, labeled dataset $L$, counter $c$.
**output:** New labeled dataset $L'$, new counter $c'$.
 1: $x \leftarrow$ independent draw from $D_{\mathcal{X}}$ (the corresponding label is hidden).
 2: **if** $x \in \text{Dis}(V)$ **then**
 3:     $L' \leftarrow L \cup \big\{(x, \text{LABEL}(x))\big\}$
 4:     $c' \leftarrow c + 1$
 5: **else**
 6:     $L' \leftarrow L \cup \big\{(x, V(x))\big\}$
 7:     $c' \leftarrow c$
 8: **end if**

---

Subroutine ERROR-CHECK checks if the version space has high error, based on item 2 of Lemma 9 – that is, if $k = k^*$, then ERROR-CHECK should never fail. Furthermore, if version space $V_i$ fails ERROR-CHECK, then $V_i$ should have small radius – see Lemma 8 for details.

---

**Algorithm 7** ERROR-CHECK

---

**input:** Version space $V \subset H_k$, labeled dataset $L$ of size $l$, confidence $\delta$.
**output:** Boolean variable $b$ indicating if $V$ has high error.
 1: Let $\delta_k := \delta/((k+1)(k+2))$ for all $k \geq 0$.
 2: $\gamma \leftarrow \min_{k' \geq k, h \in H_{k'}} \big\{ \text{err}(h, L) + 2\sqrt{\text{err}(h, L)\sigma_{k'}(l, \delta_{k'})} + 3\sigma_{k'}(l, \delta_{k'}) \big\}$
 3: **if** $\min_{h \in V} \text{err}(h, L) > \gamma + 2\sqrt{\gamma \sigma_k(l, \delta_k)} + 3\sigma_k(l, \delta_k)$ **then**
 4:     $b \leftarrow$ **true**
 5: **else**
 6:     $b \leftarrow$ **false**
 7: **end if**

---

Subroutine PRUNE-VERSION-SPACE performs update on our version space based on standard generalization error bounds. The version space never eliminates the optimal hypothesis in $H_k(S)$ when working with $H_k$. Claim 4 shows that, if at step $i$, $k = k^*$, then $h^* \in V_i$ from then on.

---

**Algorithm 8** PRUNE-VERSION-SPACE

---

**input:** Version space $V \subset H_k$, labeled dataset $L$ of size $l$, confidence $\delta$.
**output:** Pruned version space $V'$.
 1: Update version space:

$$V' \leftarrow \Big\{ h \in V : \text{err}(h, L) \leq \min_{h' \in V} \text{err}(h', L) + 2\sqrt{\text{err}(h', L)\sigma_k(l, \delta_k)} + 3\sigma_k(l, \delta_k) \Big\},$$

    where $\delta_k := \frac{\delta}{(k+1)(k+2)}$.

---

Subroutine UPGRADE-VERSION-SPACE is called when (1) a systematic mistake of the version space $V_i$ has been found by SEARCH; or (2) ERROR-CHECK detects that the error of $V_i$ is high. In either case, $k$ can be increased to the minimum level such that the updated $H_k(S)$ is nonempty. This still maintains the invariant that $k \leq k^*$.

**Algorithm 9** UPGRADE-VERSION-SPACE

**input:** Current level of hypothesis class $k$, seed set $S$, seed to be added $s$.
**output:** New level of hypothesis class $k$, new seed set $S$, updated version space $V$.
1: $S \leftarrow S \cup s$
2: $k \leftarrow \min \{k' > k : H_{k'}(S) \neq \emptyset\}$
3: $V \leftarrow H_k(S)$

### E.2  Proof of Theorem 4

This section uses the following definition of $\sigma$:

$$\sigma_k(m, \delta) = \phi(d_k, m, \delta/3) = \frac{1}{m}(d \log em^2 + \log \frac{6}{\delta}).$$

We restate Theorem 4 here for convenience.

**Theorem 7.** *There exist constants $c_1, c_2 > 0$ such that the following holds. Assume $\text{err}(h^*) = \nu$. Let $\theta_{k'}(\cdot)$ denote the disagreement coefficient of $V_i$ at the first step $i$ after which $k \geq k'$. Fix any $\epsilon, \delta \in (0, 1)$. Let $n_\epsilon = c_1 \max_{k \leq k^*} \theta_k(2\nu + 2\epsilon)(d_{k^*} \log \frac{1}{\epsilon} + \log \frac{1}{\delta})(1 + \nu^2/\epsilon^2)$ and define $C_\epsilon = 2(n_\epsilon + k^* \tau)$. Run Algorithm 2 with a nested sequence of hypotheses $\{H_k\}_{k=0}^\infty$, oracles LABEL and SEARCH, confidence parameter $\delta$, cost ratio $\tau \geq 1$, and upper bound $N = c_2(d_{k^*} \log \frac{1}{\epsilon} + \log \frac{1}{\delta})/\epsilon^2$. If the cost spent is at least $C_\epsilon$, then with probability $1 - \delta$, the current hypothesis $\tilde{h}$ has error at most $\nu + \epsilon$.*

**Remark.**  The purpose of having a bound on unlabeled examples, $N$, is rather technical— to deter the algorithm from getting into an infinite loop due to its blind self-confidence. Suppose that AA-LARCH starts with $H_0$ that has a single element $h$. Then, without such an $N$-based condition, it will incorrectly infer the labels of all the unlabeled examples drawn and end up with an infinite loop between lines 4 and 14. The condition on $N$ is very mild—any $N$ satisfying $N = \text{poly}(d_{k^*}, 1/\epsilon)$ and $N = \Omega(d_{k^*}/\epsilon^2)$ is sufficient.

*Proof of Theorem 7.* For integer $j \geq 0$, define step $j$ as the execution period in AA-LARCH when the value of $i$ is $j$.

Let $l_i = |L_i|$. Denote by $L_i^D$ the dataset containing unlabeled examples in $L_i$ labeled entirely by LABEL, i.e., $L_i^D = \{(x, \text{LABEL}(x)) : (x, y) \in L_i\}$. Note that $L_i^D$ is an iid sample from $D$.

We call dataset $L_i$ has *favorable bias*, if the following holds for any hypothesis $h$:

$$\text{err}(h, L_i^D) - \text{err}(h^*, L_i^D) \leq \text{err}(h, L_i) - \text{err}(h^*, L_i). \tag{10}$$

Let $E_i$ be the event that the following conditions hold:

1. For every $k \geq 0$, every $h \in H_k$ satisfies

$$\text{err}(h, D) \leq \text{err}(h, L_i^D) + \sqrt{\text{err}(h, L_i^D)\sigma_k(l_i, \delta_{i,k})} + \sigma_k(l_i, \delta_{i,k}),$$

$$\text{err}(h, L_i^D) \leq \text{err}(h, D) + \sqrt{\text{err}(h, D)\sigma_k(l_i, \delta_{i,k})} + \sigma_k(l_i, \delta_{i,k}).$$

   For every $h, h' \in H_k$,
$$(\text{err}(h, L_i^D) - \text{err}(h', L_i^D)) - (\text{err}(h, D) - \text{err}(h', D))$$
$$\leq \sqrt{d_{L_i^D}(h, h') \cdot \sigma_k(l_i, \delta_{i,k})} + \sigma_k(l_i, \delta_{i,k}).$$

   where $d_{L_i^D}(h, h') = \frac{1}{l_i} \sum_{(x,y) \in L_i^D} [h(x) \neq h'(x)]$, fraction of $L_i^D$ where $h$ and $h'$ disagree.

2. For every $1 \leq i' < i$, the number of LABEL queries from step $i'$ to step $i$ is at most

$$\sum_{j=i'}^i \Pr_{x \sim D_\mathcal{X}} [x \in \text{Dis}(V_{j-1})] + O\left(\sqrt{\sum_{j=i'}^i \Pr_{x \sim D_\mathcal{X}} [x \in \text{Dis}(V_{j-1})] \log(1/\delta_i)} + \log(1/\delta_i)\right),$$

   where $V_j$ denotes its final value in Algorithm 2.

Using Theorem 5 and Lemma 5, along with the union bound, $\Pr(E_i) \geq 1 - \delta_i$. Define $E := \cap_{i=1}^{\infty} E_i$, by union bound, $\Pr(E) \geq 1 - \delta$. We henceforth condition on $E$ holding.

Define

$$
\begin{aligned}
M(\nu, k^*, \epsilon, \delta, N) &:= \min \left\{ m \in \mathbb{N} : 8\sqrt{\nu \sigma_{k^*}(m, \delta_{m+k^*N,k^*})} + 35\sigma_{k^*}(m, \delta_{m+k^*N,k^*}) \leq \epsilon \right\} \\
&\leq O\left( \frac{(d_{k^*} \log(1/\epsilon) + \log(Nk^*/\delta))(\nu + \epsilon)}{\epsilon^2} \right)
\end{aligned}
$$

We say that an iteration of the loop is *verified* if Step 20 is triggered; all other iterations are *unverified*. Let $\Gamma$ be the set of $i$'s where $x_i$ gets added to the final set $L$, and $\Delta$ be the set of $i$'s where $x_i$ gets discarded. It is easy to see that if $i$ is in $\Gamma$ (resp. $\Delta$), then the $i$ is in a verified (resp. unverified) iteration.

Define $i^* := \min \left\{ i \in \Gamma : l_i \geq M(\nu, k^*, \epsilon, \delta, N) \right\}$. Denote by $k_i$ the final value of $k$ after $i$ unlabeled examples are processed.

We need to prove two claims:

1. For $i \geq i^*$, $\mathrm{err}(\tilde{h}_i) \leq \nu + \epsilon$, where $\tilde{h}_i$ is the hypothesis $\tilde{h}$ stored at the end of step $i$.

2. The total cost spent by Algorithm 2 up to step $i^*$ is at most $C_\epsilon$.

To prove the first claim, fix any $i \geq i^*$. The stored hypothesis $\tilde{h}_i$ is updated only when $i \in \Gamma$, so it suffices to consider only $i \in \Gamma$. From Lemma 10, $i \leq l_i + k^*N$. We also have $l_i \geq M(\nu, k^*, \epsilon, \delta, N)$. Since $\tilde{h}_i \in V_i$, Lemma 8 gives

$$
\begin{aligned}
\mathrm{err}(\tilde{h}_i) &\leq \nu + 8\sqrt{\nu \sigma_{k^*}(l_i, \delta_{i,k^*})} + 35\sigma_{k^*}(l_i, \delta_{i,k^*}) \\
&\leq \nu + 8\sqrt{\nu \sigma_{k^*}(l_i, \delta_{l_i+k^*N,k^*})} + 35\sigma_{k^*}(l_i, \delta_{l_i+k^*N,k^*}) \\
&\leq \nu + \epsilon,
\end{aligned}
$$

as desired.

For the second claim, we first show that for $i$ in $\Gamma$, the version space is contained in a ball of small radius (with respect to the disagreement pseudometric), thus bounding the size of its disagreement region. Lemma 8 shows that for $i \in \Gamma$, every hypothesis $h \in V_i$ has error at most $\nu + 8\sqrt{\nu \sigma_{k^*}(l_i, \delta_{i,k^*})} + 35\sigma_{k^*}(l_i, \delta_{i,k^*})$.

Thus, by the triangle inequality and Lemma 10,

$$
\begin{aligned}
V_i &\subseteq \mathrm{B}_{H_{k_i}}(h, 2\nu + 16\sqrt{\nu \sigma_{k^*}(l_i, \delta_{i,k^*})} + 70\sigma_{k^*}(l_i, \delta_{i,k^*})) \\
&\subseteq \mathrm{B}_{H_{k_i}}(h, 2\nu + 16\sqrt{\nu \sigma_{k^*}(l_i, \delta_{l_i+k^*N,k^*})} + 70\sigma_{k^*}(l_i, \delta_{l_i+k^*N,k^*})).
\end{aligned}
$$

for some $h$ in $H_{k_i}(S)$. This shows that for $i \in \Gamma$, $i \leq i^*$,

$$
\Pr_{x \sim D_{\mathcal{X}}}[x \in \mathrm{Dis}(V_i)]
$$

$$
\leq \theta_{k_i}(2\nu + 2\epsilon) \cdot \left( 2\nu + 16\sqrt{\nu \sigma_{k^*}(l_i, \delta_{l_i+k^*N,k^*})} + 70\sigma_{k^*}(l_i, \delta_{l_i+k^*N,k^*}) \right)
$$

$$
\leq \max_{k \leq k^*} \theta_k(2\nu + 2\epsilon) \cdot \left( 2\nu + 16\sqrt{\nu \sigma_{k^*}(l_i, \delta_{l_i+k^*N,k^*})} + 70\sigma_{k^*}(l_i, \delta_{l_i+k^*N,k^*}) \right), \quad (11)
$$

where the first inequality is from the definition of $\theta_{k_i}(\cdot)$ and $8\sqrt{\nu \sigma_{k^*}(l_i, \delta_{l_i+k^*N,k^*})} + 35\sigma_{k^*}(l_i, \delta_{l_i+k^*N,k^*}) \geq \epsilon$ for $i \leq i^*$, the second inequality is from $k_i \leq k^*$.

For $i \geq 1$, let $Z_i$ be the indicator of whether LABEL is queried with $x_i$ in Step 12, i.e.,

$$
Z_i = \mathbb{1}\{x_i \in \mathrm{Dis}(V_{i-1})\}.
$$

For $0 \leq k \leq k_{i^*}$, define

$i_k^0 := \min\{i \leq i^* : k_i \geq k\}$, the first step when the hypothesis class reaches $\geq k$,

$i_k := \max\{i \leq i^* : k_i \leq k\}$, the last step when the hypothesis class is still $\leq k$ by the end of that step,

$i_k' := \max\{i_k^0 \leq i \leq i_k : k_i \leq k, i \in \Gamma\}$, the last verified step for hypothesis class $\leq k$ (if exists).

We call class $k$ *skipped* if there is no step $i$ such that $k_i = k$. If level $k$ is skipped, then $i_k = i_{k-1} = i_k^0 - 1$, and $i_k'$ is undefined.

Let

$$W_k := \sum_{i=i_k^0+1}^{i_k'} Z_i$$

be the number of verified queried examples when working with hypothesis class $H_k$. Note that $W_k/\tau$ is the number of verified iterations when working with $H_k$. If level $k$ is skipped, then $W_k := 0$.

Let

$$Y_k := \sum_{i=i_k'+1}^{i_k+1} Z_i$$

be the number of unverified queried examples when working with hypothesis class $H_k$. Note that $Y_k \leq \tau$, and there is at most one unverified iteration when working with $H_k$. If level $k$ is skipped, then $Y_k := 0$.

Therefore, the total cost when working with $H_k$ is at most

$$\frac{W_k}{\tau} \cdot 2\tau + Y_k + \tau \leq 2\tau + 2W_k$$

Furthermore, Claim 4 implies that there is no unverified iteration when working with $H_{k^*}$. Hence the total cost when working with $H_{k^*}$ has a tighter upper bound, that is, $2W_{k^*}$.

As a shorthand, let $m = M(\nu, k^*, \epsilon, \delta, N)$. We now bound the total cost incurred up to time $i^*$ as

$$
\begin{aligned}
\sum_{k=0}^{k^*-1}(2\tau + 2W_k) + 2W_{k^*} &= 2\tau k^* + 2\sum_{k=0}^{k_{i^*}} W_k \\
&= 2\tau k^* + 2\sum_{k=0}^{k_{i^*}}\sum_{i=i_k^0+1}^{i_k'} Z_i \\
&= 2\tau k^* + O\left(2\sum_{i\in\Gamma: i\leq i^*} \Pr_{x\sim D_{\mathcal{X}}}[x \in \mathrm{Dis}(V_{i-1})]\right) + O\left(k^* \ln \frac{1}{\delta_{i^*}}\right) \\
&\leq 2\left(\tau k^* + O\left(\sum_{l=1}^{m-1} \max_{k\leq k^*}\theta_k(2\nu+2\epsilon)(\nu+\sigma_{k^*}(l,\delta_{l+k^*N,k^*}))\right)\right) \\
&\leq 2\left(\tau k^* + O\left(\max_{k\leq k^*}\theta_k(2\nu+2\epsilon)\sum_{l=1}^{m-1}(\nu+\sigma_{k^*}(l,\delta_{l+k^*N,k^*}))\right)\right) \\
&\leq 2\left(\tau k^* + \tilde{O}\left(\max_{k\leq k^*}\theta_k(2\nu+2\epsilon)d_{k^*}\left(1+\frac{\nu^2}{\epsilon^2}\right)\right)\right) \\
&\leq 2(\tau k^* + n_\epsilon) = C_\epsilon,
\end{aligned}
$$

where the first equality is by algebra, the second equality is from the definition of $W_k$, and the third equality is from the definition of $E$. The first inequalty is from Lemma 8, using Equation (11) to bound $\Pr_{x\sim D_{\mathcal{X}}}[x \in \mathrm{Dis}(V_{i-1})]$ and noting that $\{l_i : i \in \Gamma, i \leq i^*\} = [m]$. $\qquad\square$

Now we provide the proof of our two key lemmas(Lemmas 8 and 10).

Consider the last call of PRUNE-VERSION-SPACE in step $i$. Define $\gamma_i$ as the value of $\gamma$ in line 2 of ERROR-CHECK:

$$\gamma_i = \min_{k' \geq k_i, h \in H_{k'}} \left\{ \text{err}(h, L_i) + 2\sqrt{\text{err}(h, L_i)\sigma_{k'}(l, \delta_{i,k'})} + 3\sigma_{k'}(l, \delta_{i,k'}) \right\} \qquad (12)$$

Meanwhile, from line 1 of PRUNE-VERSION-SPACE, we have for all $h \in V_i$,

$$\text{err}(h, L_i) \leq \min_{h' \in V_i} \text{err}(h', L_i) + 2\sqrt{\text{err}(h', L_i)\sigma_{k_i}(l_i, \delta_{i,k_i})} + 3\sigma_{k_i}(l_i, \delta_{i,k_i})$$

where $V_i$ denotes its final value.

**Lemma 8.** *Assume that the following conditions hold:*

1. *The dataset $L_i$ has favorable bias, i.e. it satisfies Equation* (10)*.*

2. *The version space $V_i$ is such that* ERROR-CHECK$(V_i, L_i, \delta_i)$ *returns false, i.e. it has a low empirical error on $L_i$:*

$$\min_{h' \in V_i} \text{err}(h', L_i) \leq \gamma_i + 2\sqrt{\gamma_i \sigma_{k_i}(l_i, \delta_{i,k_i})} + 3\sigma_{k_i}(l_i, \delta_{i,k_i}). \qquad (13)$$

*Then, every $h \in V_i$ is such that*

$$\text{err}(h) \leq \nu + 8\sqrt{\nu \sigma_{k^*}(l_i, \delta_{i,k^*})} + 35\sigma_{k^*}(l_i, \delta_{i,k^*}). \qquad (14)$$

*where $V_i$ and $L_i$ denote their final values, respectively. Specifically, Equation* (14) *holds for any $h \in V_i$ such that $i \in \Gamma$ or $i + 1 \in \Gamma$.*

*Proof.* Lemma 10 shows that $k_i \leq k^*$, which we will use below.

Start with Equation (13):

$$\min_{h' \in V_i} \text{err}(h', L_i) \leq \gamma_i + 2\sqrt{\gamma_i \sigma_{k_i}(l_i, \delta_{i,k_i})} + 3\sigma_{k_i}(l_i, \delta_{i,k_i}).$$

Since $k_i \leq k^*$, $\sigma_{k_i}(l_i, \delta_{i,k_i}) \leq \sigma_{k^*}(l_i, \delta_{i,k^*})$.

From the definition of $\gamma_i$ (Equation (12)), taking $k = k^* \geq k_i$, $h = h^* \in H_{k^*}$,

$$\gamma_i \leq \text{err}(h^*, L_i) + 2\sqrt{\text{err}(h^*, L_i)\sigma_{k^*}(l_i, \delta_{i,k^*})} + 3\sigma_{k^*}(l_i, \delta_{i,k^*}).$$

Plugging the latter into the former and using $\sigma$ as a shorthand for $\sigma_{k^*}(l_i, \delta_{i,k^*})$, we have

$$\min_{h' \in V_i} \text{err}(h', L_i) \leq \text{err}(h^*, L_i) + 2\sqrt{\text{err}(h^*, L_i)\sigma} + 3\sigma + 2\sqrt{(\text{err}(h^*, L_i) + 2\sqrt{\text{err}(h^*, L_i)\sigma} + 3\sigma)\sigma} + 3\sigma$$

$$\leq \text{err}(h^*, L_i) + 2\sqrt{\text{err}(h^*, L_i)\sigma} + 6\sigma + 2(\sqrt{\text{err}(h^*, L_i)\sigma} + \sqrt{3}\sigma)$$

$$\leq \text{err}(h^*, L_i) + 4\sqrt{\text{err}(h^*, L_i)\sigma} + 10\sigma .$$

Fix any $h \in V_i$. By construction,

$$\text{err}(h, L_i) \leq \min_{h' \in V_i} \text{err}(h', L_i) + 2\sqrt{\text{err}(h', L_i)\sigma_{k_i}(l_i, \delta_{i,k_i})} + 3\sigma_{k_i}(l_i, \delta_{i,k_i}).$$

Plugging the former into the latter (recalling that $\sigma_{k_i}(l_i, \delta_{i,k_i}) \leq \sigma$) gives

$$\text{err}(h, L_i) - \text{err}(h^*, L_i) \leq 4\sqrt{\text{err}(h^*, L_i)\sigma} + 10\sigma + 2(\sqrt{\text{err}(h^*, L_i)\sigma} + \sqrt{10}\sigma) + 3\sigma$$

$$\leq 6\sqrt{\text{err}(h^*, L_i)\sigma} + 20\sigma.$$

Combined with Equation (10), we have,

$$\text{err}(h, L_i^D) - \text{err}(h^*, L_i^D) \leq 6\sqrt{\text{err}(h^*, L_i)\sigma} + 20\sigma.$$

Since $\mathrm{err}(h^*, L_i) \le \mathrm{err}(h^*, L_i^D)$,

$$\mathrm{err}(h, L_i^D) - \mathrm{err}(h^*, L_i^D) \le 6\sqrt{\mathrm{err}(h^*, L_i^D)\sigma} + 20\sigma.$$

From the definition of $E_i$,

$$\mathrm{err}(h^*, L_i^D) \le \nu + \sqrt{\nu\sigma} + \sigma.$$

$$\mathrm{err}(h) \le \mathrm{err}(h, L_i^D) + \sqrt{\mathrm{err}(h, L_i^D)\sigma} + \sigma.$$

Plugging in and simplifying algebraically gives

$$\mathrm{err}(h) \le \nu + 8\sqrt{\nu\sigma} + 35\sigma.$$

Now, if $i \in \Gamma$, the dataset $L_i$ has favorable bias from lemma 11; if $i \notin \Gamma$ and $i + 1 \in \Gamma$, the final value of $L_i$ equals some $L_j$ for some $j \in \Gamma$, therefore also has favorable bias.

Meanwhile, if $i \in \Gamma$, Algorithm 2 fails ERROR-CHECK$(V_i, L_i, \delta_i)$ for $k = k_i$. If $i \notin \Gamma$ and $i+1 \in \Gamma$, then $i + 1$ is the start of some verified iteration, i.e. $i + 1 = i_k^0$ for some $k$. Hence the final value of $V_i$ also fails ERROR-CHECK$(V_i, L_i, \delta_i)$ for $k = k_i$. In both cases, Equation (13) holds.

Therefore, if $i \in \Gamma$ or $i + 1 \in \Gamma$, then Equation (14) holds for every $h$ in $V_i$. $\qquad\square$

**Lemma 9.** *For step $i$, suppose $L_i$ has favorable bias, i.e. Equation (10) holds. Then for any $k$ and any $h \in H_k$,*

$$\mathrm{err}(h^*, L_i) - \mathrm{err}(h, L_i) \le 2\sqrt{\mathrm{err}(h, L_i)\sigma_{\bar{k}}(l_i, \delta_{i,\bar{k}})} + 3\sigma_{\bar{k}}(l_i, \delta_{i,\bar{k}}),$$

*where $\bar{k} = \max(k^*, k)$. Specifically:*

*1. for any $h \in H_{k^*}$,*

$$\mathrm{err}(h^*, L_i) - \mathrm{err}(h, L_i) \le 2\sqrt{\mathrm{err}(h, L_i)\sigma_{k^*}(l_i, \delta_{i,k^*})} + 3\sigma_{k^*}(l_i, \delta_{i,k^*}), \qquad (15)$$

*2. The empirical error of $h^*$ on $L_i$ can be bounded as follows:*

$$\mathrm{err}(h^*, L_i) \le \gamma_i + 2\sqrt{\gamma_i\sigma_{k^*}(l_i, \delta_{i,k^*})} + 3\sigma_{k^*}(l_i, \delta_{i,k^*}) \qquad (16)$$

*Proof.* Fix any $k$ and $h \in H_k$. Since $\bar{k} \ge k$, $\sigma_k(l_i, \delta_{i,k}) \le \sigma_{\bar{k}}(l_i, \delta_{i,\bar{k}})$. Similarly, $\sigma_{k^*}(l_i, \delta_{i,k^*}) \le \sigma_{\bar{k}}(l_i, \delta_{i,\bar{k}})$. Using the shorthand $\sigma := \sigma_{\bar{k}}(l_i, \delta_{i,\bar{k}})$ and noting that $h, h^* \in H_{\bar{k}}$,

$$
\begin{aligned}
\mathrm{err}(h^*, L_i) - \mathrm{err}(h, L_i) &\le \mathrm{err}(h^*, L_i^D) - \mathrm{err}(h, L_i^D) \\
&\le \sqrt{d_{L_i^D}(h^*, h) \cdot \sigma} + \sigma \\
&\le \sqrt{(\mathrm{err}(h^*, L_i) + \mathrm{err}(h, L_i)) \cdot \sigma} + \sigma. \\
&\le \sqrt{\mathrm{err}(h^*, L_i)\sigma} + \sqrt{\mathrm{err}(h, L_i)\sigma} + \sigma.
\end{aligned}
$$

where the first inequality is from Equation (10), the second inequality is from the definition of $E_i$ and the optimality of $h^*$, and the third inequality is from the triangle inequality. Letting $A = \mathrm{err}(h^*, L_i)$, $B = \mathrm{err}(h, L_i)$, and $C = B + \sqrt{B\sigma} + \sigma$, we can rewrite the above inequality as $A \le C + \sqrt{A\sigma}$. Solving the resulting quadratic equation in terms of $A$, we have $A \le C + \sigma + \sqrt{C\sigma}$, or

$$
\begin{aligned}
A &\le B + \sqrt{B\sigma} + 2\sigma + \sqrt{\sigma(B + \sqrt{B\sigma} + \sigma)} \\
&\le B + \sqrt{B\sigma} + 2\sigma + \sqrt{\sigma}(\sqrt{B} + \sqrt{\sigma}) \\
&\le B + 2\sqrt{B\sigma} + 3\sigma,
\end{aligned}
$$

or

$$\mathrm{err}(h^*, L_i) \le \mathrm{err}(h, L_i) + 2\sqrt{\mathrm{err}(h, L_i)\sigma} + 3\sigma.$$

Specifically:

1. Taking $k = k^*$, we get that Equation (15) holds for any $h \in H_{k^*}$, establishing item 1.

2. Define

$$(\hat{k}_i, \hat{h}_i) := \arg \min_{k' \geq k^*, h \in H_{k'}} \left\{ \mathrm{err}(h, L_i) + 2\sqrt{\mathrm{err}(h, L_i)\sigma_{k'}(l_i, \delta_{i,k'})} + 3\sigma_{k'}(l_i, \delta_{i,k'}) \right\}.$$

In this notation, $\gamma_i = \mathrm{err}(\hat{h}_i, L_i) + 2\sqrt{\mathrm{err}(\hat{h}_i, L_i)\sigma_{\hat{k}_i}(l_i, \delta_{i,\hat{k}_i})} + 3\sigma_{\hat{k}_i}(l_i, \delta_{i,\hat{k}_i})$. We have

$$\gamma_i + 2\sqrt{\gamma_i \sigma_{k^*}(l_i, \delta_{i,k^*})} + 3\sigma_{k^*}(l_i, \delta_{i,k^*}) \geq \mathrm{err}(\hat{h}_i, L_i) + 2\sqrt{\mathrm{err}(\hat{h}_i, L_i)\sigma_{\bar{k}}(l_i, \delta_{i,\bar{k}})} + 3\sigma_{\bar{k}}(l_i, \delta_{i,\bar{k}})$$
$$\geq \mathrm{err}(h^*, L_i),$$

where $\bar{k} = \max(k^*, \hat{k}_i)$ and the last inequality comes from applying Lemma 9 for $h' = \hat{h}_i \in H_{\hat{k}_i}$ and $\bar{k}$. This establishes Equation (16), proving item 2. $\qquad\square$

**Lemma 10.** *At any step of* AA-LARCH, $k \leq k^*$. *Consequently, for every* $i$, $i \leq l_i + k^* N$.

*Proof.* We prove the lemma in two steps.

1. Notice that there are two places where $k$ is incremented in AA-LARCH, line 6 and line 17. If $k < k^*$, neither line would increment it beyond $k^*$ as $h^* \in H_{k^*}$ and $h^*$ is consistent with $S$. If $k = k^*$, Claim 4 below shows that $k$ will stay at $k^*$. This proves the first part of the claim.

2. An iteration becomes unverified only if $k$ gets incremented, and Algorithm 2 maintains the invariant that $k_i \leq k^*$. Thus, the number of unverified iterations is at most $k^*$. In addition, each newly sampled set is of size at most $N$. So the number of unverified examples is at most $k^* N$.

   Hence, $i$—the total number of examples processed up to step $i$—equals the sum of the number of verified examples $l_i$, plus the number of unverified examples, which is at most $k^* N$. This proves the second part of the claim. $\qquad\square$

We show a technical claim used in the proof of Lemma 10 which guarantees that, on event $E$, when $k$ has reached $k^*$, it will remain $k^*$ from then on. Recall that $k_i$ is defined as the final value of $k$ at the end of step $i$; $i_k^0 = \min\{i : k_i \geq k\}$ is the step at the end of which the working hypothesis space reaches level $\geq k$.

**Claim 4.** *If* $i_{k^*}^0$ *is finite, then the following hold for all* $i \geq i_{k^*}^0$:

   *(C1)* $L_i$ *has favorable bias.*

   *(C2)* *Step* $i$ *terminates with* $k_i = k^*$.

   *(C3)* $h^* \in V_i$.

*Above,* $L_i$ *and* $V_i$ *denote their final values in* AA-LARCH.

*Proof.* By induction on $i$.

**Base Case.** Let $i = i_{k^*}^0$. Consider the execution of AA-LARCH at the start of step $i_{k^*}^0$ (line 11). Since by definition of $i_k$, the final value of $k$ at step $i_{k^*}^0 - 1$ is $< k^*$, at step $i_{k^*}^0$, line 5 or line 16 is triggered. Hence the dataset $L_{i_{k^*}^0}$ equals some verified labeled dataset $L$ stored by AA-LARCH, i.e. $L_j$ for some $j \in \Gamma$. Thus, applying Lemma 11, Claim C1 holds.

We focus on the moment in step $i = i_{k^*}^0$ when $k$ increases to $k^*$ in UPGRADE-VERSION-SPACE(line 6 or 17). Now consider the temporary $\tilde{V}_{i_{k^*}^0}$ computed in the next line (PRUNE-VERSION-SPACE). Item 2 of Lemma 9 implies that the version space $V_{i_{k^*}^0}$ is such that ERROR-CHECK($V_{i_{k^*}^0}, L_{i_{k^*}^0}, \delta_{i_{k^*}^0}$) returns false. Therefore the final value of $k$ in step $i_{k^*}^0$ is exactly $k^*$. Claim C2 follows.

Claim C2 implies the temporary $\tilde{V}_{i_{k^*}^0}$ is final. Item 1 of Lemma 9 implies that $h^* \in V_{i_{k^*}^0}$, establishing Claim C3.

**Inductive Case.** Now consider $i \geq i_{k^*}^0 + 1$. The inductive hypothesis says that Claims C1–3 hold for step $i - 1$.

Claim C1 follows from Claim C3 in step $i - 1$. Indeed, the newly added $x_i$ either comes from the agreement region of $V_{i-1}$, in which case label $y_i$ agrees with $h^*(x_i)$, or is from the disagreement region of $V_{i-1}$, in which case the inferred label $y_i$ is queried from LABEL. Following the same reasoning as the proof of Lemma 11, Claim C1 is true.

Claims C2 and C3 follows the same reasoning as the proof for the base case. $\qquad\square$

**Lemma 11.** *If $i$ is in $\Gamma$, then $L_i$ has favorable bias. That is, for any hypothesis $h$,*

$$\mathrm{err}(h, L_i^D) - \mathrm{err}(h^*, L_i^D) \leq \mathrm{err}(h, L_i) - \mathrm{err}(h^*, L_i).$$

*Proof.* We can split $L_i^D$ into two subsets, the subset where $L_i^D$ agrees with $L_i$ and the subset $Q_i^D = \{(x,y) \in L_i^D : h^*(x) \neq y\}$ where $L_i^D$ disagrees with $L_i$. On the former subset, $L_i^D$ is identical to $L_i$, thus we just need to show that

$$\mathrm{err}(h, Q_i^D) - \mathrm{err}(h^*, Q_i^D) \leq \mathrm{err}(h, Q_i) - \mathrm{err}(h^*, Q_i),$$

where $Q_i = \{(x,y) : (x,-y) \in Q_i^D\}$. Since $\mathrm{err}(h^*, Q_i^D) = 1$ and $\mathrm{err}(h^*, Q_i) = 0$, this reduces to showing that $\mathrm{err}(h, Q_i^D) \leq 1 + \mathrm{err}(h, Q_i)$, which is easily seen to hold for any $h$ as $\mathrm{err}(h, Q_i^D) \leq 1$ and $\mathrm{err}(h, Q_i) \geq 0$. $\qquad\square$