[Reviews · NeurIPS 2016]

Reviewer 1

Summary

The paper assumes an active learning setting with a sequence of nested hypothesis classes, as in SRM. For such a setting, they propose a new type of query. In contrast to simple LABEL queries their query type, called SEARCH, asks whether there is some instance in the domain that is labelled incorrectly by all hypotheses in the version space; if yes, such an instance is provided (and the learner then also knows that it has to increase the capacity of its hypothesis class). The authors show that SEARCH can simulate LABEL, but not vice versa; they argue though that it is still interesting to combine SEARCH with LABEL, since SEARCH may be more expensive than LABEL. They provide and analyze two algorithms for the realizable case, and two for the agnostic case. (However, some of these algorithms are only provided in the supplementary material.)

Qualitative Assessment

The paper is well structured and contains pretty much no typos; the authors do a good job i=of referencing related literature (as far as I can tell ... note that I am not an expert on active learning and may have missed some literature). The proposed type of queries may be of interest to the community. However, much of the presented analysis seems preliminary to me, and do not allow for conclusions strong enough to warrant publication at NIPS. For example: - what good is it to save on LABEL queries, if we cannot estimate the cost of the required SEARCH queries to achieve a reduction in LABEL queries? Can the authors provide a cost analysis that shows under which constraints on the cost of either type of query it really makes sense to use SEARCH in combination with LABEL? - a similar analysis would be needed to come to the conclusion that SEARCH may be superior to CCQ. To that end, all the authors do refer to future work in the conclusions, but I would like to see some more convincing arguments here. Another weakness is the presentation: - From my understanding of the submission instructions, the main part of the paper should include all that one needs to understand the paper (even if proofs may be in supplementary material). I thus found it awkward to have a huge algorithm listing as in Alg. 2, without any accompanying text explaining it, and to have algorithms in the supplementary material without giving at least a brief idea of the algorithms in the main body of the paper. This makes it hard to read the paper, and I think it is not appropriate for publication. - Perhaps something more should be done to convince the reader that a query of the type SEARCH is feasible in some realistic scenario. One other little thing: - what is meant by "and quantities that do appear" in line 115?

Confidence in this Review

2-Confident (read it all; understood it all reasonably well)


Reviewer 2

Summary

This article studies a type of active learning in which the algorithm has access both to the usual Label query oracle and also a "Search" oracle. The Search oracle takes a set of classifiers as input, and returns a correctly labeled example that contradicts all of them, if one exists. They first relate this oracle to others that have appeared in the literature previously, and provide several nice examples. They then propose a general active learning algorithm that uses a nested sequence of hypothesis classes, and uses the Search oracle to perform model selection, and to construct a seed data set for an active learning algorithm for each hypothesis class. They study this both in the realizable case and the agnostic case.

Qualitative Assessment

Putting aside the practicality of this framework, the theoretical analysis of the advantages of access to this type of oracle is interesting. This work reveals that the search oracle provides substantial advantages in the context of model selection. For instance, by making roughly k* search queries (where k* is the index of the model containing the Bayes classifier), Theorem 3 reveals that the number of label queries can be reduced by at least a factor of k* (and sometimes dramatically more) compared to the best known result in the prior literature on model selection in active learning (where only label queries are allowed). This can be a significant reduction in some settings. Now, on the issue of practicality, the authors make an effort to motivate each major component of the framework in terms of practical considerations. However, in many cases, these motivations seem fairly thin. A good example is the fact that, even in the agnostic case, the search oracle returns the label h*(X) of the optimal classifier h* at the returned point X, rather than the (noisy) label Y associated with that example X. The motivation given is that we can suppose the oracle will choose an example for which she is supremely confident in the label. But this motivation implicitly assumes that the only source of noise is the annotator's uncertainty about the label; this is often not the case, as factors such as the feature space representation can also introduce inherent noise (e.g., in NLP, choosing a bag-of-words representation creates noise, since the human annotator sees the words in order, and different word orders may form different valid sentences, and which of these orders is observed may affect her choice of label, so that the conditional distribution of label given the unordered list of words may have nonzero entropy). That said, I can appreciate that it is sometimes desirable for the annotator to be able to indicate a strong preference for a particular label, and supposing this strongly-preferred label to be h*(X) might be a reasonably elegant way to incorporate this extra information into the formal model, and that in some cases the h*(X) assumption might not need to strictly hold for the algorithm to have reasonable behavior in practice. Overall, I found this to be an interesting piece of work, definitely worth reading for those interested in the theory of interactive learning. However, I would expect any future works that build on it will probably tweak various aspects of the model, to weaken the search oracle, or potentially modify the type of input (or the restrictions thereon) to the oracle (e.g., I can imagine these search constraints, based on the region of agreement of a subset of H, could potentially make a minimax lower bound analysis significantly more difficult than it would be for certain natural alternatives, some of which make more practical sense and still allow for the types of regions used in the algorithms proposed here). minor comments: In the related work section, the notation d \nu/\epsilon^2 is used without being defined or explained. The main theorems in this paper are somewhat difficult to interpret in the abstract, being sample-dependent bounds on the number of labeled samples the algorithm uses. Is there any hope of obtaining concise abstract a-priori bounds, as were possible for the unions of intervals example? The wording in the discussion section is confusing to me. It seems to me that the search oracle would often be more powerful (provides more information) than the CCQ oracle, since CCQ only needs to return the noisy label while search returns the h* label. I guess the discussion is referring to the fact that CCQ can specify any region, not only a region of agreement of a subset of H (and in particular, in the realizable case this means CCQ is at least as powerful as search). So discussing these two types of queries as being a kind of trade-off in the agnostic case seems more accurate than what is currently written. Related to this, it might be interesting to discuss teasing apart the contribution of getting h*(X) labels instead of Y labels, vs the inherent power in these kinds of region-constrained mistake queries (i.e., with Y labels returned). Can one show lower bounds for region-constrained mistake queries where the Y labels are returned instead of h* labels, worse than the upper bounds proven here using the search oracle? The bibliography is inconsistent in its choice of whether or not to abbreviate author first names.

Confidence in this Review

3-Expert (read the paper in detail, know the area, quite certain of my opinion)


Reviewer 3

Summary

This paper studies active learning with access to two distinct oracles: Label and Search, where the search oracle tries to find the counterexamples for the learning process.

Qualitative Assessment

There are some crucial issues in this paper: 1. The authors may have misunderstanding about version space: the version space is the hypothesis space in which the hypothesis is consistent with the training data. It says that the optimal classifiers which has 0 error rate is in the version space. So how to find the counterexamples which the hypotheses in the version space classify incorrectly? 2. Finding the counterexamples is very different from labeling the informative examples. For example, in standard active learning, the example which the hypotheses in the version space classify most differently is the informative example; while in the Search Oracle, the example which the hypotheses classify INCORRECTLY is the informative example. It is hard to know which example is classified INCORRECTLY by the hypothesis before we query many examples’ label. So, if the Search Oracle can tell a counterexample, this example will dramatically improve the classifier, but it is not practicable in general active learning setting.

Confidence in this Review

2-Confident (read it all; understood it all reasonably well)


Reviewer 4

Summary

This paper consider the active learning setting. It proposes an algorithm based on the two oracles LABEL and SEARCH and that can offer significant improvement over LABEL.

Qualitative Assessment

General comments: - The presentation is clear but hard to follow for a reader not used to active learning literature. - The theoretical results seem sound. - In introduction, the problem is motivated by class imbalance for ML classification task. It would be nice to demonstrate the relevance of the algorithm by showing some comparison on a classification task.

Confidence in this Review

1-Less confident (might not have understood significant parts)


Reviewer 5

Summary

This paper investigates an active learning scenario where the learner is endowed with an oracle called "search" which provides counter examples to a particular set of hypothesis (if possible). The (stated) motivation for introducing this oracle is to allay the "rare class" problem. In some situations with class imbalance, traditional active learning algorithms exhaust their budget before observing even one example of a particular class. The paper treats a nested sequence of hypothesis classes, increasing in complexity. The most basic algorithm proposed starts with the simplest hypothesis class, then upon finding a counter example to the current set of consistent hypotheses, increases complexity. The paper then presents three more complicated algorithms, one (presented in the supplementary materials) is a refined version of the basic algorithm. The second (also in supplementary materials) is agnostic and the last is agnostic and "any time" and enforces a maximum amount of data to be labeled.

Qualitative Assessment

Strengths: Proposes an oracle with easy to understand motivation which has not been adequately treated in literature Treats not only the realizable case but and agnostic and "any time" settings, giving label complexity bounds for each Provides helpful examples to understand situations where search & label beats label alone. Weaknesses: Two of the algorithms are relegated to the supplementary materials, though their corresponding sample complexity results are stated in the main text. This causes a loss of context and makes it difficult to understand the results without looking at the appendix.

Confidence in this Review

1-Less confident (might not have understood significant parts)